



**Near East Desertification: impact of Dead Sea drying on convective rainfall**

Samiro Khodayar[1,2] and Johannes Hoerner[1]

[1]Institute of Meteorology and Climate Research (IMK-TRO), Karlsruhe Institute of

 Technology (KIT), Karlsruhe, Germany

[2]Mediterranean Centre for Environmental Studies (CEAM), Valencia, Spain

Submitted to Atmospheric Chemistry and Physics

(HyMeX Inter-journal SI)

* Corresponding author. E-mail address: Khodayar_sam@gva.es (S. Khodayar)

Mediterranean Centre for Environmental Studies (CEAM),

Technological Park, Charles R. Darwin Street, 14 46980 - Paterna - Valencia - Spain





**Abstract**
The Dead Sea desertification-threatened region is affected by continual lake level decline
and occasional, but life-endangering flash-floods. Climate change has aggravated such
issues in the past decades. In this study, the impact of the Dead Sea drying on the severe
convection generating heavy precipitation in the region is investigated. Perturbation
simulations with the high-resolution convection-permitting regional climate model
COSMO-CLM and several numerical weather prediction (NWP) runs on an event time
scale are performed over the Dead Sea area.  A reference simulation covering the 2003
to 2013 period and a twin sensitivity experiment, in which the Dead Sea is dried out and
set to bare soil, are compared. NWP simulations focus on heavy precipitation events
exhibiting relevant differences between the reference and the sensitivity decadal
realization to assess the impact on the underlying convection-related processes.
On a decadal scale, the difference between the simulations points out that in future
regional climate, under ongoing lake level decline, a decrease in evaporation, higher air
temperatures and less precipitation is to expect.  Particularly, an increase in the number
of dry days and in the intensity of heavy precipitation is foreseen. The drying of the Dead
Sea is seen to affect the atmospheric conditions leading to convection in two ways: (a)
the local decrease in evaporation reduces moisture availability in the lower boundary
layer locally and in the neighbouring, directly affecting atmospheric stability. Weaker
updrafts characterize the drier and more stable atmosphere of the simulations where the
Dead Sea has been dried out. (b) Thermally driven wind system circulations and resulting
divergence/convergence fields are altered preventing in many occasions convection
initiation because of the omission of convergence lines.

*Key Words: Dead Sea drying, climate change, convection, heavy precipitation,*

*boundary layer,* wind systems, *high-resolution modelling*





## 1. Introduction

The Eastern Mediterranean and the Middle East is a sensitive climate change area (Smiatek et al. 2011). The anticipated warming in the 21$^{st}$ century combined with the general drying tendency, suggest important regional impacts of climate change, which should be investigated to assess and mitigate local effects on society and ecosystems. The Dead Sea basin, dominated by semi-arid and arid climates except by the north-western part that is governed by Mediterranean climate (Greenbaum et al. 2006), is an ideal area to study climate variation in the Near East. It was already discussed by Ashbel (1939) the influence of the Dead Sea on the climate of its neighbouring regions. The change in the climate of the Dead Sea basin caused by the drying of the Dead Sea has also been evidenced in the last decades (Alpert et al. 1997; Cohen and Stanhill 1996; Stanhill 1994). The Dead Sea is the lowest body of water in the world (~ -430 m) surrounded by the Judean Mountains (up to ~ 1 km amsl) to the west and to the east by the Maob Mountains (up to ~ 3 km amsl). The area in between is rocky desert. The complex topography of the area favours the combined occurrence of several wind regimes in addition to the general synoptic systems, namely valley and slope winds, Mediterranean breezes and local lake breezes (e.g. Shafir and Alpert 2011). These wind systems are of great importance for the living conditions in the region since they influence the visibility and the air quality (e.g. Kalthoff et al. 2000; Corsmeier et al. 2005) as well as the atmospheric temperature and humidity. Since the Dead Sea is a terminal lake of the Dead Sea Valley, no natural outflow exists, being evaporation the main loss of water (Metzger et al. 2017). Through the high evaporation the lake level declines and results in a desertification of the shoreline and a changing fraction of water and land surface in the valley. The documented Dead Sea water level drop of about 1 m/y in the last decades (Gavrieli et al. 2005) severely affects agriculture, industry and the environmental conditions in the area, thus, leading to substantial economic losses (Arkin and Gilat 2000).

The Jordan River catchment and Dead Sea-exhibit in the north, annual precipitation in the order of 600-800 mm, whereas in the south, there is an all year arid climate with an annual precipitation of <150 mm (Schaedler and Sasse 2006). Rain occurs between October and May and can be localized or widespread (Dayan and Sharon 1980) with annual variations of the same order of magnitude as the rainfall itself (Sharon and Kutiel 1986). Rainfall varies seasonally and annually, and it is often concentrated in intense showers (Greenbaum et al. 2006). The Dead Sea basin is prone to flash flooding caused mainly by severe convection generating heavy precipitation (Dayan and Morin 2006). Flash floods are among the most dangerous meteorological hazards affecting the


Mediterranean countries (Llasat et al 2010), thus, knowledge about the processes
shaping these events is of high value. This is particularly relevant in arid climates, where
rainfall is scarce, and often, local and highly variable. In flood-producing rainstorms,
atmospheric processes often act in concert at several scales. Synoptic-scale processes
transport and redistribute the excess sensible and latent heat accumulated over the
region and subsynoptic scale processes determine initiation of severe convection and
the resulting spatio-temporal rainfall characteristics.  The main responsible synoptic
weather patterns leading to heavy rainfall in the region are in general well known and
described in previous publications (e.g. Belachsen et al. 2017; Dayan and Morin 2006).
Belachsen et al. (2017) pointed out that three main synoptic patterns are associated to
these rain events: Cyprus low accounting for 30% of the events, Low to the east of the
study region for 44%, and Active Read Sea Trough for 26%. The first two originate from
the Mediterranean Sea, while the third is an extension of the Africa monsoon. Houze
(2012) showed that orographic effects lead to enhanced rainfall generation; rain cells are
larger where topography is higher. Sub-synoptic scale processes play a decisive role in
deep convection generation in the region. Convection generated by static instability
seems to play a more important role than synoptic-scale vertical motions (Dayan and
Morin 2006). The moisture for developing intensive convection over the Dead Sea region
can be originated from the adjacent Mediterranean Sea (Alpert und Shay-EL 1994) and
from distant upwind sources (Dayan and Morin 2006).
In this study, the climatic change at the Dead Sea region caused by its drying is
investigated focusing on the impact on atmospheric conditions leading to heavy
precipitating convection in the region. The relevance of the Dead Sea as a local source
of moisture for precipitating convection as well as the impact of the energy balance
partitioning changes and related processes caused by the drying of the Dead Sea are
investigated. With this purpose, a sensitivity experiment with the high-resolution regional
climate model COSMO-CLM [Consortium for Small scale Modelling model (COSMO)-in
Climate Mode (CLM); Böhm et al. 2006] is conducted. The high horizontal grid spacing
used (~ 2.8 km) resolves relevant orographic and small-scale features of the Dead Sea
basin, which is not the case when coarser resolution simulations are performed.
Moreover, at this resolution convection is explicitly resolved instead of being
parametrized, which has been already extensively demonstrated to be highly beneficial
for the simulation of heavy precipitation and convection-related processes (e.g., Prein et
al., 2013; Fosser et al., 2014; Ban et al., 2014). This effort, to the knowledge of the
authors, has not been previously attempted in the region.



The impact of completely drying the Dead Sea on the regional atmospheric conditions
and precipitating convection is discussed. A decadal simulation and several event-based
Numerical Weather Prediction (NWP) runs covering the eastern Mediterranean are
carried out. A process understanding methodology is applied to improve our knowledge
about how sub-synoptic scale processes leading to severe convection are affected by
the drying of the Dead Sea. The article is organized as follows. Section 2 provides an
overview of the data and the methodology used. Then, in section 3, the climatology of
the region based on the high-resolution convection-permitting decadal simulation is
presented and the impact of drying the Dead Sea is examined across scales. Finally,
conclusions are discussed in section 4.

**2. Data and methodology**
**2.1 The COSMO-CLM model**
In this investigation, the regional climate model (RCM) of the non-hydrostatic COSMO
model, COSMO-CLM, is used (Version 5.0.1). It has been developed by the Consortium
for Small-scale modeling (COSMO) and the Climate Limited-area Modeling Community
(CLM) (Böhm et al., 2006). It uses a rotated geographical grid and a terrain-following
vertical coordinate. The model domain covers the southern half of the Levant, centered
around the Dead Sea, with a horizontal resolution of 7 km and 2.8 km, 60 vertical levels
and a time step of 60 and 20 seconds, respectively. The driving data for the 7 km with a
horizontal resolution of 0.25° is derived from the IFS (Integrated Forecasting System)
analysis, the spectral weather model of ECMWF (European Centre for Medium-Range
Weather Forecast). Additionally, orography data from GLOBE (Global Land One-km
Base Elevation Project) of NOAA (National Oceanic and Atmospheric Administration)
and soil data from HWSD (Harmonized Worlds Soil Database) TERRA is used. HWSD
is a global harmonization of multiple regional soil data sets with a spatial resolution of
0.008° (FAO, 2009), resulting in 9 different soil types in the model, namely 'ice and
glacier', 'rock / lithosols', 'sand', 'sandy loam', 'loam', 'loamy clay', 'clay', 'histosols', and
'water'.
With a horizontal resolution below 3 km, convection can be resolved directly (Doms and
Baldauf, 2015). The model physics includes a cloud physics parameterization with 5
types of hydrometeors (water vapor, cloud water, precipitation water, cloud ice,
precipitation ice), a radiative transfer scheme based on a delta-two-stream solution



(Doms et al., 2011) and a roughness-length dependent surface flux formulation based
on modified Businger relations (Businger et al., 1971).
Multiple model runs have been performed. A 7 km run from 2003 to 2013 with daily output
is used as nesting for two 2.8 km runs over the same time span. The Dead Sea is dried
out and replaced with soil types from the surrounding area in one of them (SEN), the
other one is used as reference (CLIM). For the detailed investigation of convective events
on 14.11.2011 and 19.11.2011, sub-seasonal simulations have been performed with the
same settings as the decadal simulation, but with hourly outputs.
**2.2 Methodology**
In order to assess the impact of the drying of the Dead Sea on the atmospheric conditions
leading to severe convection in the region, a set of sensitivity experiments was
performed. A decadal simulation covering the 2003 to 2013 time period was carried out
with the convection permitting 2.8 km COSMO-CLM model. Lateral boundary conditions
and initial conditions are derived from the European Centre for Medium-Range Weather
Forecasts (ECMWF) reanalysis data. The COSMO-CLM 7 km is used as nesting step in
between the forcing data and the 2.8 km run. This reference simulation will be hereafter
referred to as REF$^{CLIM}$ simulation. Parallel to this, a sensitivity experiment (hereafter
SEN$^{CLIM}$) is carried out in which the Dead Sea is dried out and set to bare soil on -405 m
level (depth of the Dead Sea in the external data set, GLOBE (Hastings and Dunbar,
1999)). After examination of the results, the first year of simulations is considered spin-
up time, thus, our analysis covers the 2004-2013 period.
Regional dry and wet periods are identified and quantified in the simulations by means
of the Effective Drought Index (EDI; Byun and Wilhite 1999; Byun and Kim 2010). The
EDI is an intensive measure that considers daily water accumulations with a weighting
function for time passage normalizing accumulated precipitation. The values are
accumulated at different time scales and converted to standard deviations with respect
to the average values. Here we use an accumulation period of 365 days. EDI dry and
wet periods are categorized as follows: moderate dry periods $-1.5 < EDI < -1$, severe dry
periods $-2 < EDI < -1.5$, and extreme dry periods $EDI < -2$. Normal periods are revealed by
$-1 < EDI < 1$ values.
Based on daily mean values, precipitation and evapotranspiration distribution and
possible trends in the 10-year period are assessed. The study area is divided in four
subdomains centred at the Dead Sea to examine dependencies in relation to the regional
patterns (Figure 1). Differences in the annual cycle and temporal evolution of





precipitation and evapotranspiration between the REF<sup>CLIM</sup> and SEN<sup>CLIM</sup> are discussed.
Also, differences in the near-surface and boundary layer conditions and geopotential
height patterns are examined. Geographical patterns of mean evapotranspiration and
precipitation and differences with respect to the reference simulation are assessed.
Probability distribution functions (PDFs), and the Structure, Amplitude and Location
(SAL: Wernli et al. 2008) analysis methodologies are used to illustrate differences in the
mean and extreme precipitation between the reference and the sensitivity experiments.
The SAL is an object-based rainfall verification method. This index provides a quality
measure for the verification of quantitative precipitation forecasts considering three
relevant aspects of precipitation pattern: the structure (S), the amplitude (A), and the
location (L). The A component measures the relative deviation of the domain-averaged
rainfall; positive values indicate an overestimation of total precipitation, negative values
an underestimation. The component L provides an estimation of the 'accuracy of
location', comparing the proportion of high and low rainfall totals within each object. The
component S is constructed in such a way that positive values occur if precipitation
objects are too large and/or too flat and negative values if the objects are too small and/or
too peaked, quantifying the physical distance between the centres of mass of the two
rainfall fields to be compared. Perfect agreement between prediction and reference are
characterized by zero values for all components of SAL. Values for the amplitude and
structure are in the range (−2, 2), where ±0.66 represents a factor of 2 error. The location
component ranges from 0 to 2, where larger values indicate a greater separation
between centres of mass of the two rainfall fields. This is done by selecting a threshold
value of 1/15 of the maximum rainfall accumulation in the domain (following Wernli et al.
2008). The structure and location components are thus independent of the total rainfall
in the domain.

Differences in the temporal evolution of precipitation between the REF<sup>CLIM</sup> and SEN<sup>CLIM</sup>
are identified. Those events in which an area-mean (study area, Figure 1) difference
between both simulations higher than ±0.1 mm/d exists are classified attending to their
synoptic scale environment, and atmospheric stability conditions.
Although Dayan and Morin (2006) discuss that in general large-scale vertical motions do
not provide the sufficient lifting necessary to initiate convection, it was demonstrated by
Dayan and Sharon (1980) that a relationship exists between the synoptic-scale weather
systems and deep moist convection, being those systems responsible for the
moisturizing and destabilization of the atmosphere prior to convective initiation. They
pointed out that indices of instability proved the most efficient determinants of the



environment characterizing each rainfall type in the region. Thus, two indicators of the atmospheric degree of stability/instability, namely the Convective Available Potential Energy (CAPE; Moncrieff and Miller 1976) and the KO-index (Andersson et al. 1989), are examined in this study. The CAPE is a widely known index indicating the degree of conditional instability. Whereas, the KO-index, which is estimated based on the equivalent potential temperature at 500, 700, 850 and 1000 hPa (following the recommendations by Bolton 1980), describes the potential of deep convection to occur as a consequence of large-scale forcing (Andersson et al. 1989; Khodayar et al. 2013). Generally, regions with KO-index < 2 K and large-scale lifting are identified as favourable for deep convection. Parcel theory (50 hPa ML (Mixed Layer) parcel) and virtual temperature correction (Doswell and Rasmussen 1994) are applied to these calculations.

Based on the above criteria, a separation was made between events with widespread rainfall and those more localized. Among the latter, we selected two events to illustrate the local impacts on the boundary layer conducive to deep moist convection. Particularly, differences in the amount, structure and location of precipitation are assessed by examining the spatial patterns and the SAL verification method. High-resolution simulations with the NWP COSMO 2.8 km model are performed with hourly output temporal resolution and covering a 3-day period (including 48-h prior to the day of the event, from 00 UTC) to capture atmospheric pre-conditions conducive to deep moist convection. For this, a reference simulation, REF$^{NWP}$, and a sensitivity experiment, SEN$^{NWP}$, are carried out for each event.

## 3. Results and discussion

### 3.1 Climatology of the Dead Sea region

*Annual cycle*

To assess the climatology of the study region (Figure 1) the annual evaporation and precipitation cycles based on daily means of the respective quantities are investigated (Figure 2). Additionally, we examine the evolution of specific humidity ($Qv_{2m}$) and temperature at 2 m ($T_{2m}$) as well as total column integrated water vapour (IWV) and low-boundary layer (< 900 hPa) equivalent potential temperature ($\Theta_e$). Possible changes in the atmospheric stability conditions are evaluated by examination of the CAPE and KO-index. In Figure 2, all grid points over the study region (Figure 1) and the time period





2004-2013 are considered. Differences between the REF$^{CLIM}$ and the SEN$^{CLIM}$
simulations are also discussed.
The annual cycle of evaporation shows minimum values in the autumn season (around
October, ~ 0.1 mm/d) and maximum evaporation in spring (around March, ~ 0.4 mm/d).
The dependency with the precipitation cycle is clear with maximum values of the latter
around March and rain occurring between October and May (Figure 2a) in agreement
with observations in the area (Dayan and Sharon 1980). The difference between the
evaporation in the REF$^{CLIM}$ and the SEN$^{CLIM}$ simulations indicates a mean decrease in
the order of 0.02 (February) to ~ 0.1 (August) mm/d in the absence of the Dead Sea
water (SEN$^{CLIM}$). The largest difference is in the dry period (May to October) when water
availability is less dependent on precipitation, and evaporation is higher over the Dead
Sea in contrast to the minimum values over land (Metzger et al. 2017). In general, there
is a decrease of about 0.5 % in precipitation in the "non-Sea" simulation, SEN$^{CLIM}$. In
contrast to the differences in evaporation, precipitation differences between the
reference and the sensitivity experiment occur in both directions during the rain period,
from October to May. Examining the total number over the whole decadal simulation it is
seen that the number of dry or wet days (> 0.1 mm/d) or heavy precipitation events is
not largely affected in the sensitivity experiment. In general, the number of dry days
increases (fewer wet days) in the SEN$^{CLIM}$ simulation, whereas the number of high
intensity events show almost no variation. For each simulation, the difference between
precipitation and evaporation is negative mainly in spring and summer contributing to the
dryness in the region. Furthermore, the negative difference between the REF$^{CLIM}$ and
SEN$^{CLIM}$ simulations indicates that the PREC-EVAP difference is higher in the SEN$^{CLIM}$
simulation probably in relation to the reduced evaporation over the dry sea area and the
general decrease in the precipitation amount in the region.
In addition to the reduced evaporation and precipitation in the whole domain in the
SEN$^{CLIM}$ simulation a drier and warmer lower-troposphere is identified (Figure 2b) in
agreement with the observational assessment by Metzger et al. (2017) of the cooling
effect of evaporation on air temperature in the region. The annual cycle of IWV and
$\Theta_{e<900hPa}$ in Figure 2c show that the impact of the dry Dead Sea resulting evaporation is
less pronounced when a deeper atmospheric layer is considered. Indeed, $\Theta_{e<900hPa}$
evolution evidences that the warming effect due to the decreased evaporation in the
SEN$^{CLIM}$ simulation is restricted to the near surface.
In Figure 2d, the annual cycle of areal mean CAPE displays larger values in the period
from August to November, being this the period more favourable for convection. Positive



CAPE differences between the REF$^{CLIM}$ and the SEN$^{CLIM}$ simulations are presumably in
relation to the identified distinct lower-atmospheric conditions, being these more
favourable and consequently CAPE values higher in the REF$^{CLIM}$ simulation. In the same
period, the KO-index indicates a more potentially unstable atmosphere, i.e. prone to
deep convection because of large-scale forcing, and larger differences between
simulations.
To further assess the most affected areas in our investigation domain, the study region
is divided in four subdomains surrounding the Dead Sea (Figure 3). Annual cycles are
separately investigated to take into consideration the relevant differences in orography,
soil types, and distance to the coast among others (Figure1), which are known to have
a significant impact in the precipitation distribution in the region (e.g. Belachsen 2017;
Houze 2012). In agreement with the well-known precipitation distribution in the region
most of the events occur in A1 (north-east) and A2 (north-east). Also, in these
subdomains larger differences between the REF$^{CLIM}$ and SEN$^{CLIM}$ simulations are
identified pointing out the relevance of the Dead Sea evaporation in the pre-convective
environment for rainfall episodes over the study area (Figure3a). Considering only land
grid points almost no difference between simulations is found in the evaporation annual
cycle of A1 and A2 (Figure3b) suggesting the distinct amount of moisture advected
towards A1 and A2 from the Dead Sea in REF$^{CLIM}$ and SEN$^{CLIM}$ as responsible for the
differences in the boundary layer conditions conducive to convection. Also, in these
subdomains the dryer and warmer lower boundary layer and the reduced instability in
the SEN$^{CLIM}$ are recognized
*Inter-annual variability*
In Figures 4 we discuss the inter-annual variability (based on monthly-daily areal mean
values) of evaporation, precipitation as well as drought evolution.
The reduced evaporation in the annual cycle of the SEN$^{CLIM}$ simulation for the whole
investigation domain, resulting from the drying of the Dead Sea and affected evaporation,
remains from year to year (Figure 4a). Larger differences between the simulations occur
in the May to November months in agreement with the annual cycle in Figure 2a. This,
and the time period of the maximum/minimum is constant over the years. A tendency
towards lower evaporation at each simulation and higher differences between both at
the end of the period are identified. An inter-annual fluctuation is observed in both
REF$^{CLIM}$ and SEN$^{CLIM}$ simulations. The yearly rate of evaporation shows, for example, in
REF$^{CLIM}$ maximum values of about 7 mm in 2011 and around 17 mm in 2012. This is in
agreement with the positive correlation expected between precipitation and evaporation,





a trend towards decreased precipitation and a correspondence between drier years such
as the 2011-2012 period and lower annual evaporation are seen in Figure 4b. Year to
year EDI calculations in Figure 4c help us identify the regional extreme dry and wet
periods. The EDI range of variation from about -1 to 2 for the whole period of simulation
indicates that the dry condition is the common environment in the area, while the wet
periods, EDI up to 6, could be identified as extreme wet periods (relative to the area), in
this case in the form of heavy precipitation events. Maximum positive EDI values are in
the first months of the year in agreement with the precipitation annual cycle in Figure 2,
whereas minimal EDI values occur in summer and autumn indicative of the dry conditions
in these periods. Differences in the EDI calculations from both simulations reveal distinct
precipitation evolutions and denote timing differences in the occurrence of the
precipitation events. When the regional climate evolution is examined in combination
with the impact on the number of heavy precipitation events (Table 1) the impact is
stronger in the dry period of 2011 (Figure 4a). About six events show relevant differences
in this period, contrary to the average 3 episodes per year.
*Spatial distribution*
The geographical patterns of evaporation and precipitation are presented in Figure 5.
Over the Dead Sea, the simulated average annual evaporation for the period under
consideration is in the order of 1500-1800 mm/y, in contrast to the values in the deserts
east and south, where the evaporation is less than 20 mm/y. Observed annual
evaporation of this lake is known to be about 1500 mm and to vary with the salinity at
the surface of the lake and freshening by the water inflow (Dayan and Morin 2006). Over
land, higher evaporation is seen over the Judean Mountains and the Jordanian
Highlands. High correlation with the orography and soil types is seen (Figure 1).
Particularly, in the Jordanian Highlands where maximum evaporation is around 200
mm/y, the complex topography coincides with sandy loam soils, whereas most of the soil
in study region is defined as loamy clay or clay (Figure 1). The evaporative difference
field between simulations in Figure 5a shows a highly inhomogeneous patchiness not
evidencing any relationship with orography or soil type, but rather with changes in the
precipitation pattern in the SEN$^{CLIM}$ simulation as seen in Figure 5b.
In agreement with the temporal series of areal mean precipitation in Figure 3 higher
annual precipitation are in the north-west and -east, with respect to the southern regions.
Topographic features exert a large impact on precipitation distribution with maxima of
about 175 to 300 mm/y over the Judean Mountains and the Jordanian Highlands. To the
northern end of the Dead Sea valley, the largest precipitation difference between the


REF$^{CLIM}$ and the SEN$^{CLIM}$ simulations is identified, rather than directly over the Dead Sea
area noting the importance of advected moisture from the Dead Sea evaporative flux
upslope and along the Dead Sea valley as well as the indirect effects of a different spatial
distribution of low-tropospheric water vapour in the occurrence of precipitating
convection.
Regarding the impact on the large-scale conditions, differences in the spatial pattern and
strength of the 500 hPa geopotential height field are identified over the Dead Sea (not
shown). In the 10-year mean, differences up to 0.002gpdm higher in SEN than in REF
are observed. Around the Dead Sea area, the differences are smaller and more irregular.
Generally, the differences are higher in the east of the Dead Sea than in the west.

*Precipitation probability distribution function*
While the probability for lower intensity precipitation is very similar in the REF$^{CLIM}$ and the
SEN$^{CLIM}$ simulations differences are recognized in the higher precipitation intensities,
from about 150 mm/d (Figure 6a). Particularly, above 180 mm/d extreme precipitation
values occur less frequent at the SEN$^{CLIM}$ simulation where a drier, warmer and more
stable atmosphere is identified (Figure 2).
*SAL*
The use of the SAL method in this study differs from the approach frequently presented
in literature since it is here not our purpose to examine differences between the simulated
field and observations (adequate observations for this comparison are not available in
the area), but to compare changes regarding the structure, amount and location of the
precipitation field between our reference and sensitivity experiments.  Figure 6b shows
that when the mean precipitation over the whole simulation period is considered all three
SAL components are close to zero 0, meaning that very small differences are found.
However, when single precipitation events in the REF$^{CLIM}$ simulation are compared with
the same period at the SEN$^{CLIM}$ simulation, larger differences regarding structure,
amount and location of rainfall events are found. For further examination of this issue
two exemplary heavy precipitation events (indicated by boxes in Figure 6b) are analysed
in detail. In both cases, a negative A-component is recognized, that is, less precipitation
falls in the SEN$^{CLIM}$ simulation. The S-component also evidences the change in the
structure of the convective cells. The L-component is low meaning that the convective
location does not change significantly in the SEN$^{CLIM}$ simulation, in contrast to the
intensity and structure of the cells.




## 3.2 Sensitivity of atmospheric conditions to the Dead Sea drying: episodic investigation

Among those events exhibiting differences in the precipitation field between both simulations (Table 1 and Figure 6b) two situations occurring in the time period of the 14 to 19 November 2011 are investigated in the following.

In this term, the synoptic situation is characterized by a Cyprus low and its frontal system located over the Dead Sea at about 00 UTC on the 15 November 2011 and at 12 UTC on the 18 November 2011. The low-pressure system and its frontal system induced strong south-westerly to westerly winds with mean wind velocities up to 15 m/s.

In the first situation (hereafter CASE1), in association with the western movement of the cold front a convective system develops over the Jordanian Highlands with precipitation starting at about 21 UTC on the 14 November 2011. This convective system is of high interest because of the large difference in its development between the REF[14.11] and the SEN[14.11] simulations.

In Figure 7a the 24-h accumulated precipitation, from 14.11 09 UTC to 15.11 08 UTC, in the investigation area is shown for the REF[14.11] and the SEN[14.11] simulations. Two precipitation areas are seen, on the north-western and north-eastern of the Dead Sea. Larger difference between models is on the north-eastern region (24-h accumulated precipitation > 100 mm/d in REF[14.11], while < 50 mm/d in SEN[14.11]), which is the focus of our analysis.

The REF[14.11] simulation shows that in the 6 hours period prior to the initiation of convection the pre-convective atmosphere and more specifically the lower boundary layer exhibit a moist (IWV ~ 24-30 mm, qvPBLmax ~ 7-10 g/kg) and unstable (CAPE ~ 1100 J/kg; KO-index ~ -8 K; not shown) air mass on the western side of the investigation area, particularly close to the western Mediterranean coast, and drier (IWV~ 8-16; qvPBLmax ~ 4-6 g/kg) and more stable conditions (CAPE< 200 J/kg; KO-index ~ 0-2 K) on the eastern side of the domain (Figure 7b). A maximum gradient of about 5 g/kg from west to east is established in the lower boundary layer.

Main differences between both simulations are over the Dead Sea (IWV difference up to 2 mm and qvPBL up to 1.5 g/kg) and north and north-east of it, but almost similar conditions everywhere else. In our target area (subdomain of investigation where the



convection episode takes place (red box in Figure 7)), north-east of the Dead Sea, a
drier and a more stable atmosphere is identified at the SEN[14.11] simulation.
The evolution of the wind circulation systems in the area is similar in both simulations
(Figure 7c). The 700 hPa, 850 and 950 hPa winds dominantly blow from the south south-
west during the pre-convective environment advecting the moist unstable air mass
towards the Dead Sea valley and north-east of it, directly affecting the atmospheric
conditions at the target area. In both simulations, the passage of the cold front over the
Dead Sea establishes a strong southerly wind from about 10 UTC on the 14 November
419    2011.

Prior to this time, dry air was advected below about 850 hPa towards the target area
from the east. The turning of the low-level winds and the resulting moistening of the
atmosphere is well and equally captured by both simulations (Figure 8a). Furthermore,
at the near-surface, from about 16 UTC, ~ 5 h prior to convection initiation in the target
area, a near-surface convergence line forms at the foothills of the northern part of the
Jordanian Highlands, which is also well and equally captured by both simulations (Figure
8b). The lifting provided by the convergence line triggers convection in the area.
However, the drier and more stable atmosphere in the SEN[14.11]simulation results in less
intense convection, weaker updrafts, and reduced precipitation at the eastern slope of
the valley.

In the second case, CASE2, we address an episode of localized convection taking place
on the north-western edge of the Dead Sea in the REF simulation, whereas no
convection develops in the SEN simulation. The isolated convection in the REF
simulation left about 50 mm rain in 3 h starting at about 03 UTC on the 19 November
2011 (Figure 9).
In contrast to CASE1, the modification of the pre-convective environment relevant for
convective initiation is in this case dominated by dynamical changes in the mesoscale
circulations. Differences in the evolution and strength of the Mediterranean Sea Breeze
(MSB), the Dead Sea breeze and orographic winds influence atmospheric conditions in
the target area leading to the assistance to or to the absence of convection. The most
significant difference observed between the simulations is in the development of a strong
near-surface convergence line in the REF simulation (which is not present in the SEN
simulation hindering convection in the area), which forms about 2 h before convective
initiation (Figure 10).



Even in the first hours of the 18 November 2011 differences in the speed and direction
of the near-surface winds over the Dead Sea and on the eastern flank of the Jordanian
Highlands could be identified. A fundamental difference between simulations occurs from
about 17 UTC when strong westerly winds indicating the arrival of the MSB reach the
western shore of the Dead Sea. One hour later, in the REF[19.11] run the MSB strongly
penetrates the Dead Sea valley reaching as far as the eastern coast in the centre to
south areas. However, in the SEN[19.11] simulation the MSB does not penetrate downward,
instead strong northerly winds flow along the valley (Figure 10a). Numerous
observational and numerical studies carried out to investigate the dynamics of the MSB
(e.g. Naor et al. 2017; Vuellers et al. 2018) showed that the downward penetration of the
MSB results from temperature differences between the valley air mass, which is warmer
than the maritime air mass. An examination of temperature differences along a near-
surface north-south valley transect (positions in Figure 10a) indicates a decrease of
about 4 ºC at the near-surface over the dried Dead Sea area in contrast to negligible
changes on a parallel transect inland, on the western coast of the Dead Sea. These
evidences the notorious impact of the absence of water in the valley temperature, thus,
gradients in the region. The colder valley temperatures do not favour the downward
penetration of the MSB, which strongly affects the atmospheric conditions in the valley.
Moreover, a north-easterly land breeze is visible from about 20 UTC on the eastern shore
of the Dead Sea in the REF[19.11] simulation, but not in the SEN[19.11] simulation (Figure
10b). This situation reflects an interesting case different from the ones generally
presented in former investigations in the area (e.g. Alpert et al. 1997 ; and Alpert et al.
2006b) in which due to the recent weakening of the Dead-Sea breeze, mainly because
of the drying and shrinking of the Sea, the Mediterranean breeze penetrates stronger
and earlier into the Dead-Sea Valley increasing the evaporation because of the strong,
hot and dry wind.
Mountain downslope winds develop in both simulations from about 22 UTC. One hour
later, strong northerly valley flow in the northern part of the Dead Sea contrasts with the
westerly flow in the SEN[19.11] simulation (Figure 10c). As the valley cools down during
night time in the SEN simulation, T2m decreases about 1 K from 20 UTC to 03 UTC in
contrast with the 0.1 K decrease of the Dead Sea in the REF simulation, the temperature
gradient weakens and the northerly valley flow present in the REF simulation is absent
in the SEN simulation. During the night, the synoptic conditions gain more influence than
the local wind systems governing the conditions in the valley during day time. South-
easterly winds prevail in the valley in both simulations. Much stronger wind velocities are





reached in the REF simulation, confirming the sensitivity of large-scale dynamics to near-
surface climate change-induced impacts.
The encounter of the north north-westerly and south south-easterly winds over the Dead
Sea area in the REF[19.11] simulation induces the formation of a convergence zone, which
intensifies and extends offshore over the next hours and determines the location of
convective initiation. Meanwhile, homogeneous south-easterly winds are observed in the
SEN simulation (Figure 10d).
The differences in the wind circulations contribute to a different distribution of the
atmospheric conditions in the target area, particularly, low-tropospheric water vapour as
seen in the vertical cross sections in Figure 11. The evolution of the atmospheric
conditions in the 3-h period prior to convective initiation evidences the deeper and wetter
boundary layer in the REF[19.11] simulation at the north-western foothills of the ridge at the
Jordanian Highlands. Differences of IWV up to 2 mm, and of instability (CAPE) close to
200 J/kg are found in this area (not shown). This is the location of the convergence line
where convective updrafts, which start close to the ground, are triggered reaching a
maximum vertical velocity of about 5 m/s above the convergence zone in the REF[19.11]
simulation.

**4. Conclusions**
The drying and shrinking of the Dead Sea has been extensively investigated in the last
decades from different points of view. This process has been related to significant local
climate changes which affect the Dead Sea valley and neighboring regions. The climate
of the Dead Sea is very hot and dry. But occasionally the Dead Sea basin is affected by
severe convection generating heavy precipitation, which could lead to devastating flash
floods.
In this study, high-resolution COSMO model simulations are used to assess the impact
of the Dead Sea on the occurrence of convective precipitation in the region. A set of high-
resolution, ~ 2.8 km, climate simulations covering the period 2003 to 2013, and several
numerical weather prediction (NWP) runs on an event time scale (~ 48-36 h) are
performed over the Dead Sea area. On a decadal time scale, two simulations are carried
out. The first "reference" run with the Dead Sea area, and a second run "sensitivity" in
which the Dead Sea is dried out and set to bare soil. The NWP simulations focus on two
heavy precipitation events exhibiting relevant differences between the reference and the
sensitivity decadal runs. A total of four simulations are performed in this case.



As the energy balance partitioning of the Earth's surface changes due to the drying of
the Dead Sea, relevant impacts could be identified in the region. From a climatological
point of view, in a future regional climate under ongoing Dead Sea level decline, less
evaporation, higher air temperatures and less precipitation is to expect. Reduced
evaporation over the Dead Sea occurs from May to October. The cooling effect of
evaporation in the neighboring areas results in an increase of T-2m in the absence of
the Dead Sea. Atmospheric conditions, such as air temperature and humidity, are mostly
affected in the lower-tropospheric levels, which in turn influence atmospheric stability
conditions, hence, precipitating convection. The number of dry days is reduced, but in
general the number of dry/wet days is not largely affected by the drying of the Dead Sea;
rather the structure and intensity of the heavier precipitation events is changed. While a
general and homogeneous decrease in evaporation is seen at the SEN$^{CLIM}$ simulation,
precipitation deviations occur in both directions, which could suggest and impact on the
timing of the events. A relevant year to year variability is observed in evaporation-
precipitation which indicates the need of long time series of observations to understand
local conditions and to validate model simulations.
The detailed analysis of two heavy precipitation events allowed us to further assess the
possible causes and the processes involved regarding the decrease in precipitation
intensity or the total omission of convection with respect to the reference simulation in
the absence of the Dead Sea water. Two main components, strongly affected by the
drying of the Dead Sea, are found to be highly relevant for the understanding of the
environmental processes in the Dead Sea region.
(a) First, the lower-atmospheric boundary layer conditions. Changes in the energy
balance affect the atmosphere through the heat exchange and moisture supply. The
drying of the Dead Sea in the SEN simulations and the resulting decrease in local
evaporation, impact the Dead Sea Basin conditions and the neighbouring areas. A
reduction in boundary layer humidity and an increase in temperature result in a general
decrease of atmospheric instability and weaker updrafts indicating reduced deep-
convective activity. Main differences on the atmospheric conditions are directly over the
Dead Sea, but these conditions are frequently advected to neighbouring areas by the
thermally driven wind systems in the region which play a key role for the redistribution of
these conditions and the initiation of convection.
(b) Secondly, wind systems in the valley. In the arid region of the Dead Sea Basin with
varied topography, thermally and dynamically driven wind systems are key features of
the local climate. Three different scales of climatic phenomena coexist: The



Mediterranean Sea Breeze (MSB), the Dead Sea breeze and the orographic winds,
valley-, and slope-winds, which are known to temper the climate in the Dead Sea valley
(Shafir and Alpert, 2011). The drying of the Dead Sea in the SEN simulation disturbs the
Dead Sea thermally driven wind circulations. The Dead Sea breezes are missing, weaker
wind speeds characterize the region and along valley winds are consequently affected.
Furthermore, the dynamics of the Mediterranean breeze penetration into the Jordan
Valley are affected.
Consequently, the impacts on convection initiation and development are twofold:
(i) Distinct redistribution of atmospheric conditions, locally or remotely, which yields to
different atmospheric conditions that in the absence of the Dead Sea result in a reduced
moisture availability in the lower atmospheric levels and increased stability hindering
convection or reducing the intensity of the events.
(ii) Modification of the divergence/convergence field. The absence of the Dead Sea
substantially modifies the wind circulation systems over the Dead Sea valley, which leads
to the omission of convergence lines which act as triggering mechanism for convection.
We can conclude that in general the lack of sufficient low-atmospheric moisture in
relation to the drying of the Dead Sea, the increase of atmospheric stability in addition to
an absence or reduction in the intensity of the convergence zones, works against
initiation or intensification of precipitating convection in the area. The relevance of the
small-scale variability of moisture and the correct definition and location of convergence
lines for an accurate representation of convective initiation illustrates the limitation and
the lack of adequate observational networks in the area and the need for high-resolution
model simulations of boundary layer processes to predict intense and localised
convection in the region.
These results contribute to gain a better understanding of expected conditions in the
Dead Sea valley and neighbouring areas under continual lake level decline. Energy
balance partitioning and wind circulation systems are determinant for local climatic
conditions, e.g. temperature and humidity fields as well as aerosol redistribution,
therefore, any change should be well understood and properly represented in model
simulations of the region. In a further step, the authors will assess the impact of model
grid resolution on the horizontal and vertical flow field in the region across scales,
including the impact on large-scale dynamics. We will also put emphasis in trying to
better understand the dynamics of the MSB under lake level decline using high-resolution
modelling, especially the contrasting behaviour pointed out in this study. Fine resolution



simulations up to 100 m will be performed for this purpose. Furthermore, we will provide
a verification of the complex chain of processes in the area using unique measurements
in the framework of the interdisciplinary virtual institute Dead Sea Research VEnue
(DESERVE; Kottmeier et al., 2016).

**Author contribution**
SK wrote the manuscript, analysed the data, interpreted the results and supervised the
work. JH carried out data analysis, interpretation of results and prepared all the figures.

**Acknowledgements**
The first author's research was supported by the Bundesministerium für Bildung und
Forschung (BMBF; German Federal Ministry of Education and Research). The authors
acknowledge the colleagues at the Karlsruhe Institute of Technology (KIT) involved in
the interdisciplinary virtual institute Dead Sea Research VEnue (DESERVE) for their
support and interesting discussions. We acknowledge Sebastian Helgert and Alberto
Caldas Alvarez for their assistance in the preparation of the simulations. This article is a
contribution to the HyMeX program.












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





**Tables**

| | PREC diffmn | REF PMX | SEN PMX | Synoptic Situation | REF CAPEmx | SEN CAPEmx | REF KOmn | SEN KOmn | Localised/ Widespread (Subarea affected) |
|---|---|---|---|---|---|---|---|---|---|
| 09.12.2004 | -0,10 | 30,09 | 31,31 | ARST | 1 | 1 | 4,85 | 4,85 | W (A1, A2) |
| 14.01.2006 | 0,11 | 45,64 | 54,64 | Cyprus Low | 239 | 225 | 6,57 | 6,54 | L/W (A1, A3) |
| 17.04.2006 | -0,11 | 57,41 | 56,09 | Syrian Low | 43 | 47 | 1,97 | 1,94 | L (A1, A4) |
| 11.04.2007 | -0,29 | 42,61 | 70,20 | Cyprus Low | 686 | 679 | -4,77 | -4,70 | L (A2, A4) |
| 14.04.2007 | -0,12 | 134,36 | 127,79 | Cyprus Low | 573 | 576 | -1,95 | -1,92 | L (A1, A2, A3, A4) |
| 13.05.2007 | 0,16 | 41,82 | 47,90 | Syrian Low | 436 | 81 | -5,30 | -5,29 | L (A1, A2) |
| 28.01.2008 | 0,14 | 23,11 | 17,24 | Syrian Low | 7 | 7 | 5,12 | 5,12 | W (A1, A3) |
| 26.10.2008 | 0,23 | 139,01 | 125,73 | ARST | 1274 | 1361 | -5,50 | -4,08 | L (A3) |
| 14.11.2008 | -0,30 | 40,83 | 45,55 | ARST | 25 | 7 | 1,37 | 1,38 | L (A2, A4) |
| 15.05.2009 | 0,39 | 59,28 | 68,84 | Syrian Low | 433 | 429 | -3,90 | -3,91 | L (A1, A2, A3, A4) |
| 16.05.2009 | -0,20 | 49,23 | 42,28 | Syrian Low | 208 | 203 | -2,30 | -2,36 | L (A1, A2, A3) |
| 01.11.2009 | 0,19 | 166,21 | 111,79 | Cyprus Low | 435 | 445 | -5,03 | -4,46 | L (A1, A2) |
| 16.01.2011 | -0,11 | 73,02 | 72,03 | Syrian Low | 49 | 37 | 7,82 | 7,83 | L/W (A1, A4) |
| 29.05.2011 | 0,24 | 44,51 | 32,73 | Cyprus Low | 158 | 170 | -10,27 | -10,26 | W (A2) |
| 16.11.2011 | 0,11 | 42,65 | 9,34 | Cyprus Low | 2 | 0 | -7,14 | -7,12 | L (A1, A2) |
| 18.11.2011 | -0,11 | 90,07 | 93,04 | Cyprus Low | 386 | 304 | -9,14 | -9,16 | L (A1) |
| 19.11.2011 | 0,11 | 28,68 | 34,69 | Cyprus Low | 356 | 378 | -8,61 | -8,65 | L (A1) |
| 20.11.2011 | -0,03 | 58,11 | 12,36 | Cyprus Low | 133 | 81 | -7,60 | -7,46 | L (A2, A4) |
| 23.10.2012 | -0,20 | 29,88 | 41,64 | ARST | 2068 | 2097 | -5,83 | -5,59 | L (A1, A2) |
| 10.11.2012 | 0,11 | 27,20 | 22,56 | Cyprus Low | 218 | 215 | 3,97 | 3,98 | W (A1) |
| 24.11.2012 | 0,21 | 155,77 | 117,81 | ARST | 189 | 286 | -2,18 | -1,95 | L (A1, A2, A3) |
| 26.11.2012 | 0,11 | 41,48 | 54,33 | ARST | 354 | 332 | 4,19 | 4,37 | L (A3, A4) |



**Table 1:** Classification of heavy precipitation cases in the decadal simulation covering
the period 2004 to 2013. The areal-mean (study area, Figure 1) difference ($PREC_{diffmn}$)
and maximum grid precipitation in the reference ($REF_{PMX}$) and sensitivity ($SEN_{PMX}$)
realizations, the synoptic situation, and the stability conditions illustrated by maximum
grid point CAPE ($CAPEmx$) and minimum grid point KO-index ($KOmn$) are summarized.
Additionally, the nature of the precipitation, localized (L) or widespread (W) and the main
subarea affected (following division in Figure 1; A1, A2, A3, A4) are listed.













**Figures**

(a)

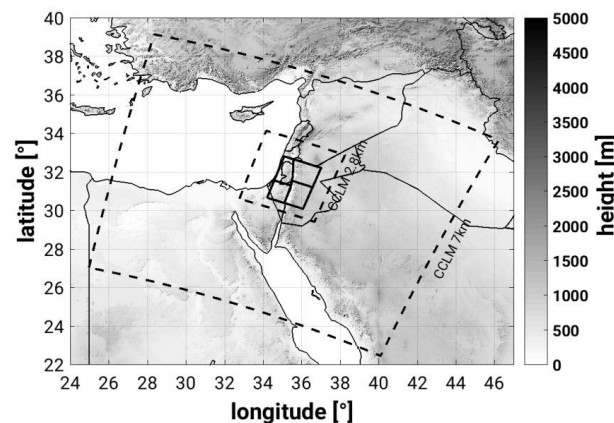


(b)

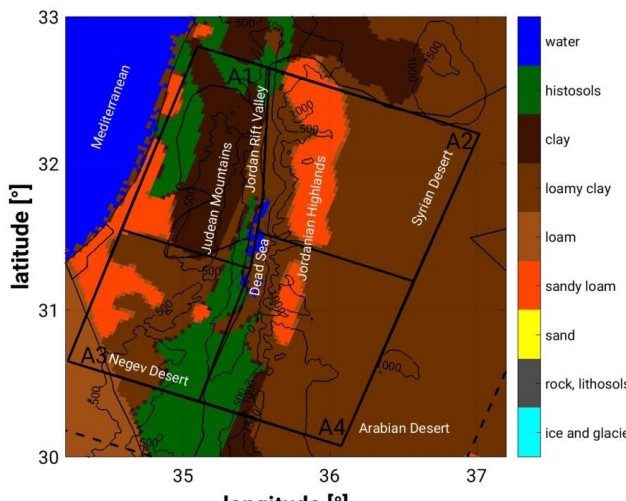



Figure 1: (a) Topography (m above msl), simulation domains (dashed lines) and study
area (bold line). (b) Model soil types (colour scale), topography (black isolines) and study
area (black bold line) including the 4 subdomains to be examined, A1-4 (Area 1-4).





(a)










(b)





(c)


(d)





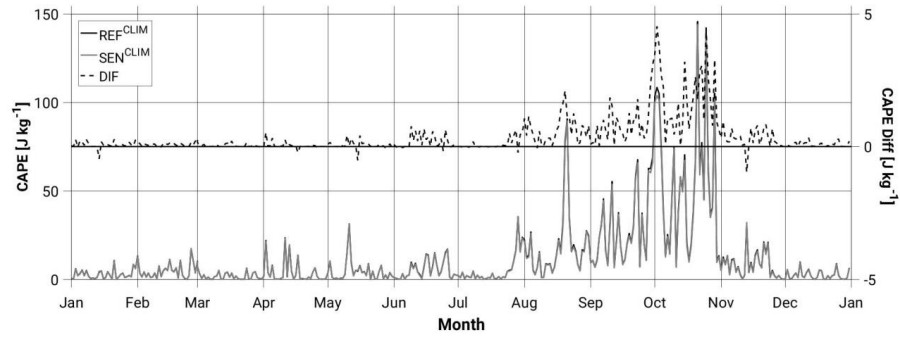





Figure 2: Annual cycle of the areal-daily averaged (and differences (black dashed line))
of (a) evaporation, precipitation, and precipitation minus evaporation (b) specific humidity
and temperature at 2-m, and (c) $\Theta_e$ below 950 hPa and IWV, and (d) CAPE and KO-
index, from the REF$^{CLIM}$ (full black line) and the SEN$^{CLIM}$ (full grey line) simulations. All
grid points in the investigation domain (Figure 1) and the period 2004 to 2013 are
considered.








(a)




Area1 (NW)                               Area2 (NE)

Area3 (SW)                               Area4 (SE)

(b)
Area1 (NW)                               Area2 (NE)










Figure 3: Annual cycle of the areal-daily averaged (and differences (black dashed line))
of (a) precipitation for areas A1, A2, A3, A4 (see Figure 1b), and (b) evaporation, specific
humidity and temperature at 2-m, and CAPE for areas A1 and A2, from the REF[CLIM] (full
black line) and the SEN[CLIM] (full grey line) simulations. Only land points in the
investigation domain (Figure 1) and the period 2004 to 2013 are considered.





























(a)

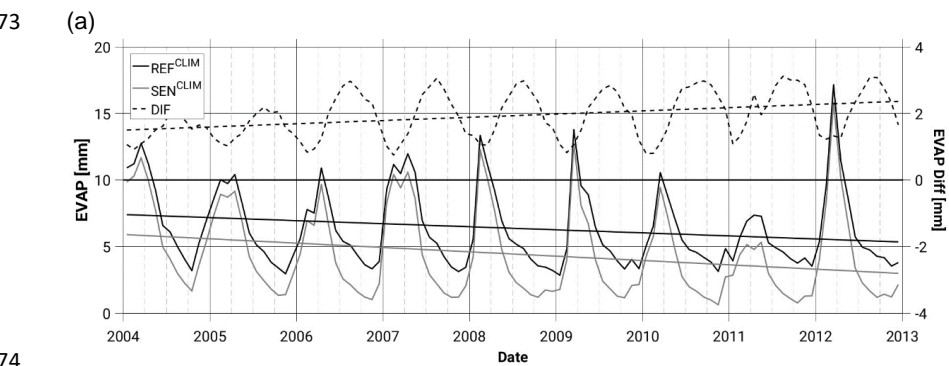


(b)

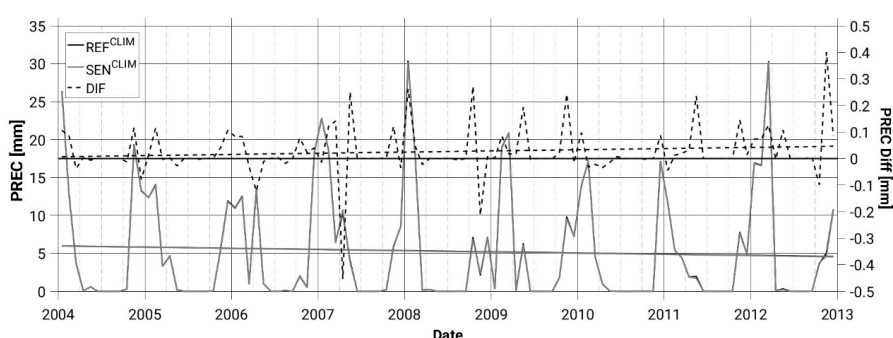


(c)

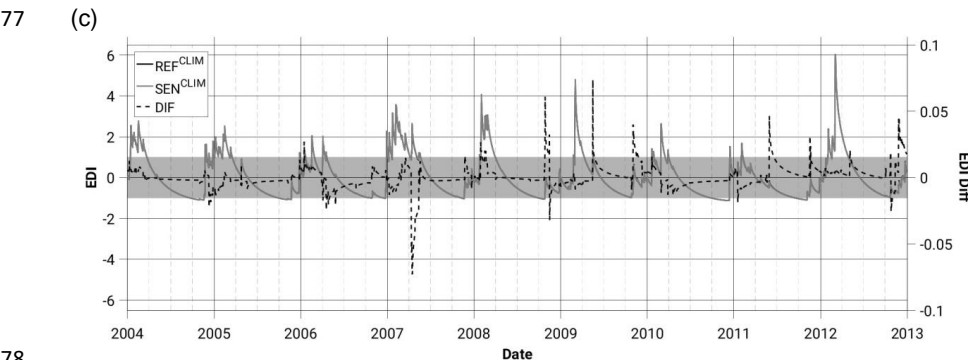



Figure 4: Temporal evolution of the monthly-daily areal mean values of (a) Evaporation,
(b) Precipitation, (c) Effective Drought Index (EDI), from the REF$^{CLIM}$ (full black line) and
the SEN$^{CLIM}$ (full grey line) simulations. Differences are depicted with black dashed lines.
The light grey band in (c) indicates the common soil state (-1<EDI<+1). All grid points in
the investigation domain (Figure 1) and the period 2004 to 2013 are considered.

(a)



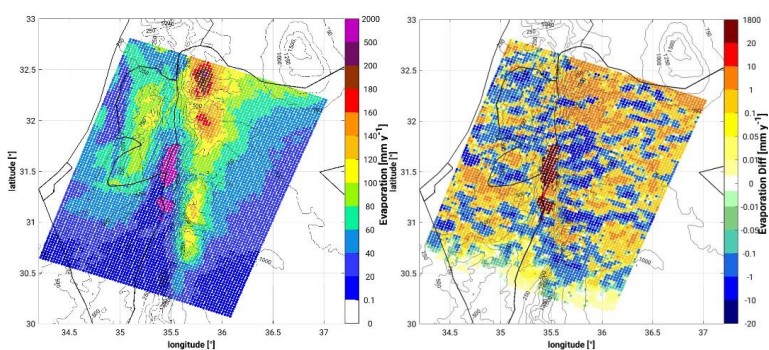



(b)

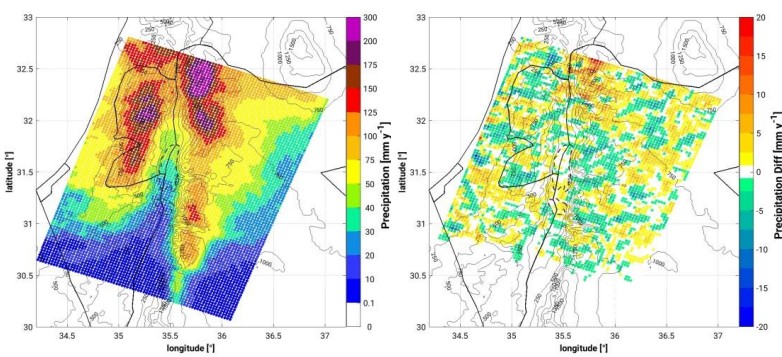



Figure 5: Spatial distribution of (a) evaporation in the REF[CLIM] simulation (left) and the
difference between the REF[CLIM] and SEN[CLIM] simulations (right), and (b) precipitation in
the REF[CLIM] simulation (left) and the difference between the REF[CLIM] and SEN[CLIM]
simulations (right). The period 2004 to 2013 is considered.











(a)                                    (b)

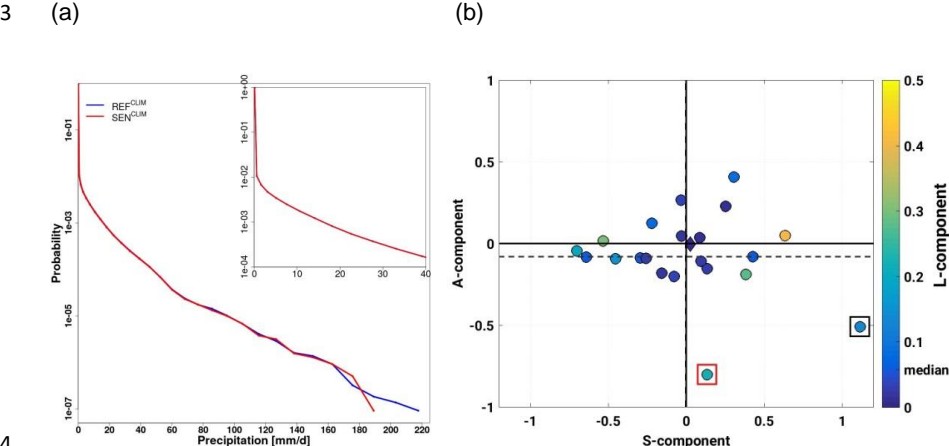

Figure 6: (a) Probability density function of daily precipitation intensities. All grid points in the investigation domain (Figure 1) and the period 2004 to 2013 are considered. (b) SAL diagram between REF$^{CLIM}$ and SEN$^{CLIM}$ simulations. Every circle corresponds to a simulated heavy precipitation event (listed in Table 1). The diamond (close to the zero-zero) illustrates the mean of all events. A-component (amplitude), S-component (structure), L-component (location). The inner colour indicates the L-component. Squares point out the two events examined in this study, CASE1 and CASE2 (see section 3.2).




(a)

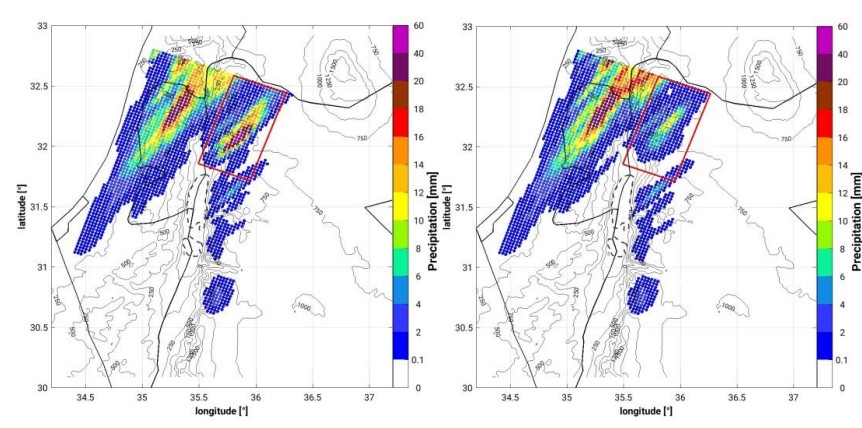


(b)

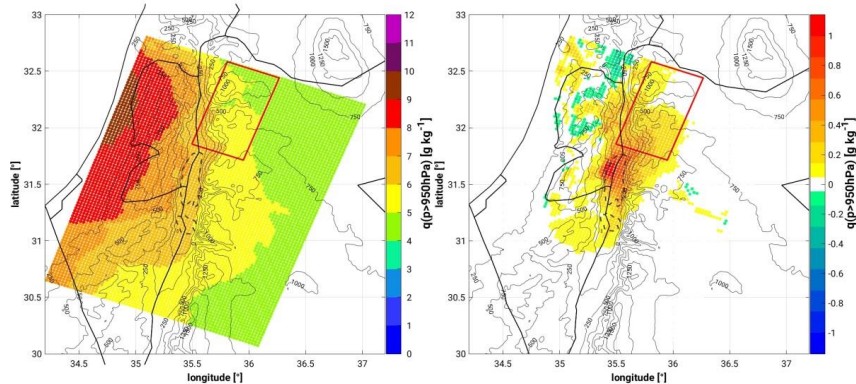














(c)

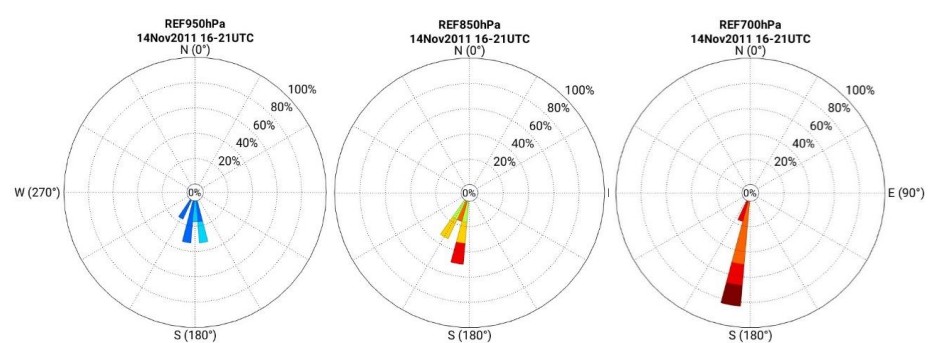



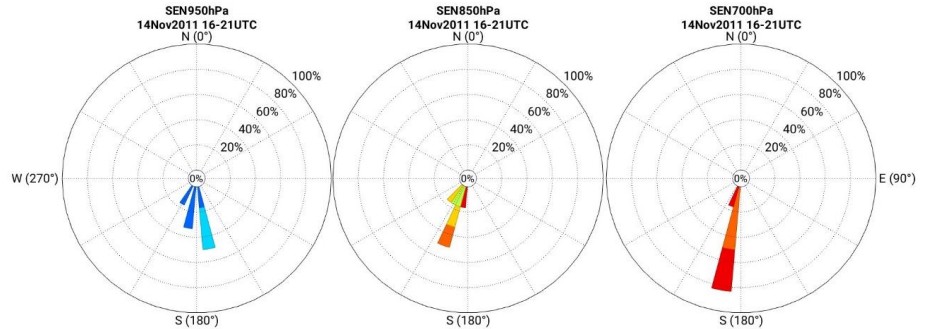


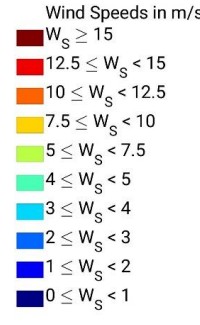










Figure 7: Spatial distribution of 24-h mean from 14.11 09 UTC to 15.11 08 UTC of, (a)
precipitation and (b) specific humidity below 950 hPa, from the REF[14.11] simulation (left)
and the difference between the REF[14.11] and SEN[14.11] simulations, as a mean for the 6-h
period prior to convection initiation in the target area (14 November 16 UTC to 21 UTC),
and (c) wind conditions at 700 hPa, 850 hPa, and 950 hPa (no relevant differences with
respect to the 10-m field) for the same time period. Wind roses are centred at about
35.82°E-32.07°N in our target area.


























(a)

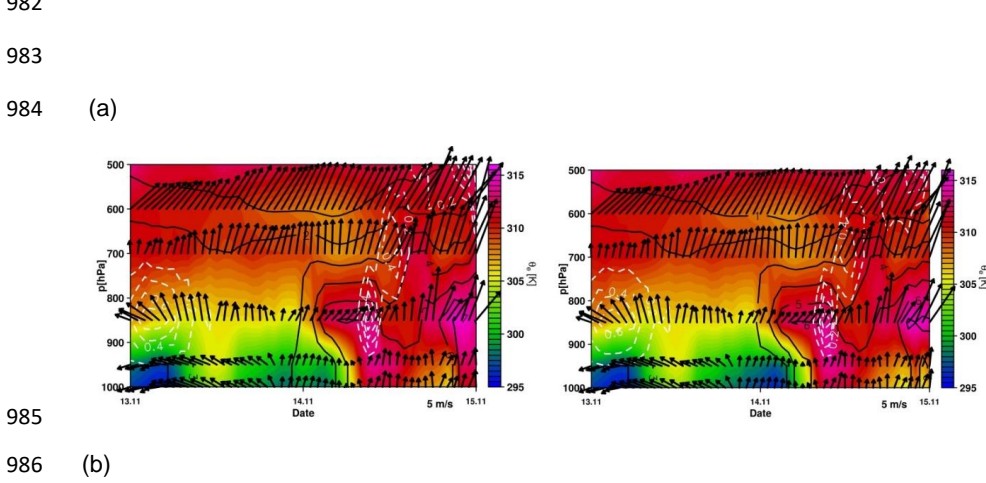


(b)

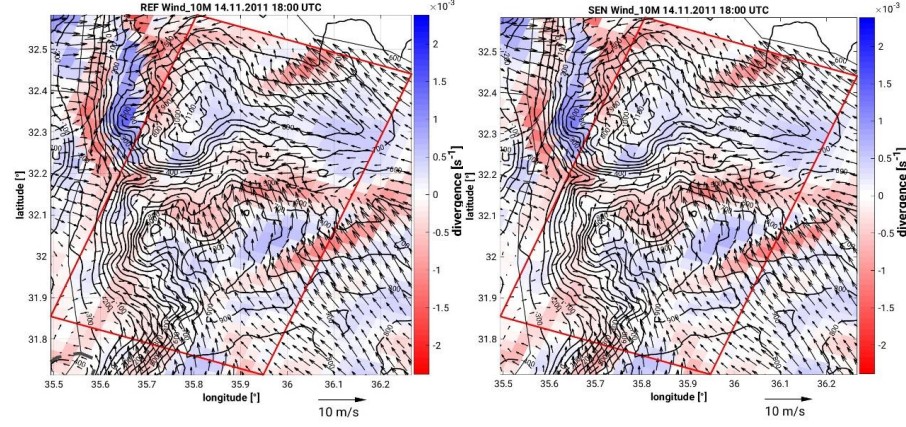



Figure 8: (a) Vertical-temporal cross-section of equivalent potential temperature (colour
scale; K), specific humidity (black isolines; g/kg), horizontal wind vectors (north-pointing
upwards, m/s) and vertical velocity (white dashed contours with 0.1 m/s increments) of
the REF[14.11] (left) and SEN[14.11] (right) simulations, over a representative grid point in the
sub-study region, 32.05°N 35.79°E. (b) Spatial distribution of 10-m horizontal wind (wind
vectors; m/s) and corresponding divergence/convergence field (colour scale; s⁻¹) at 18
UTC on the 14 November 2011 from the REF[14.11] (left) and SEN[14.11] (right) simulations.







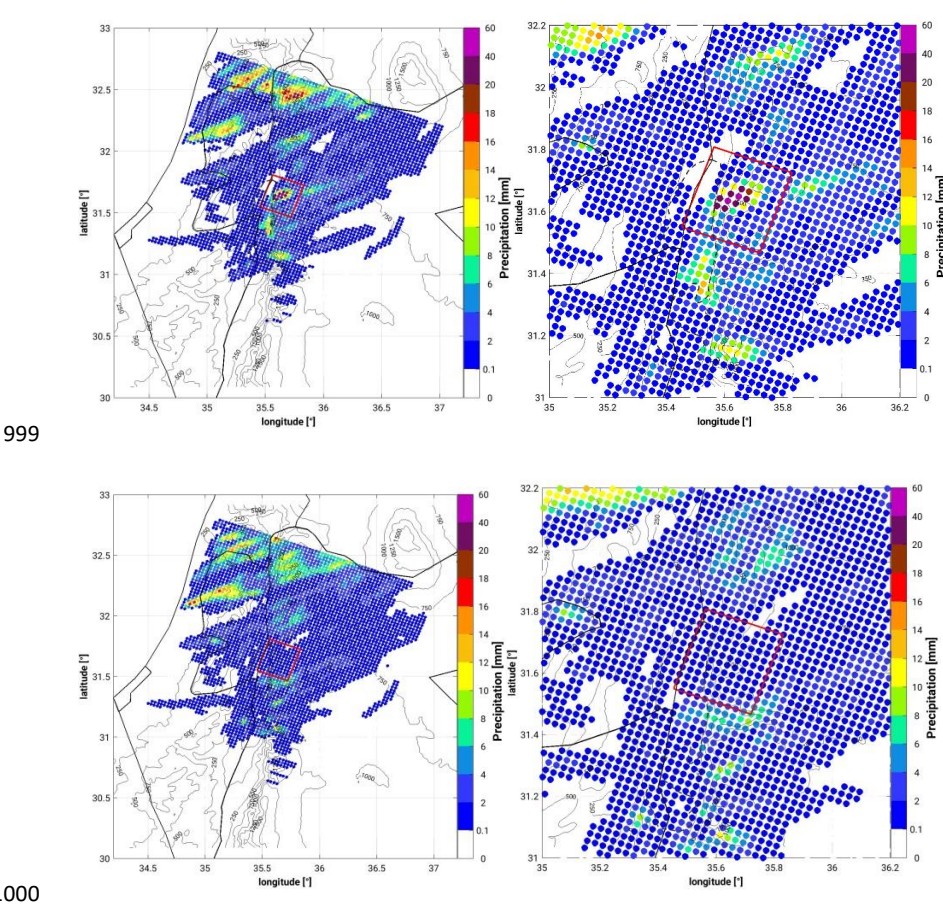


Figure 9: 24-h mean spatial distribution of precipitation from the REF[19.11] simulation (top-left; zoom top-right) and the SEN[14.11] simulation (bottom-left; zoom bottom-right) for the period 18 November 2011 11 UTC to 19 November 2011 10 UTC.














(a)

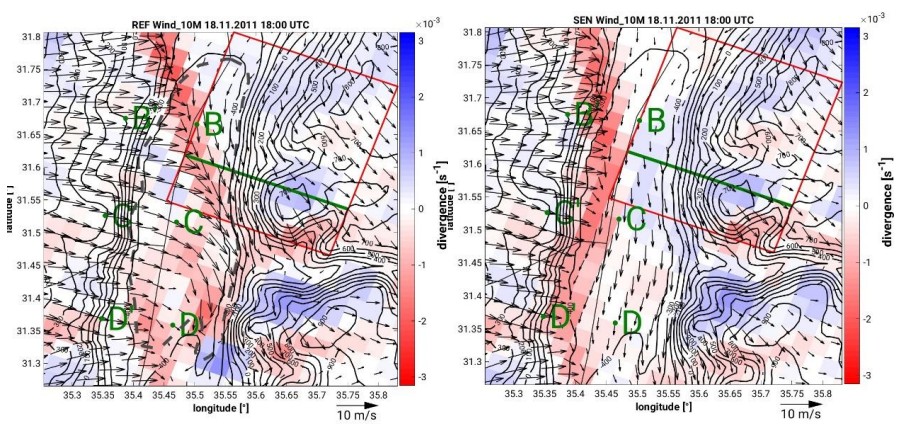


(b)

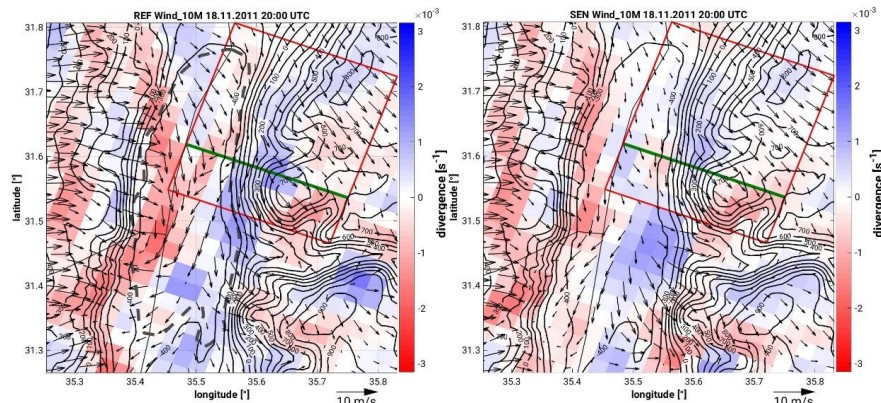












(c)

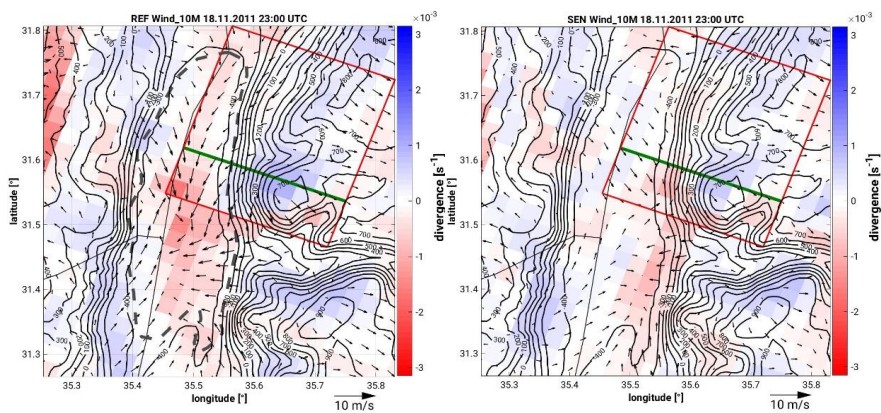



(d)

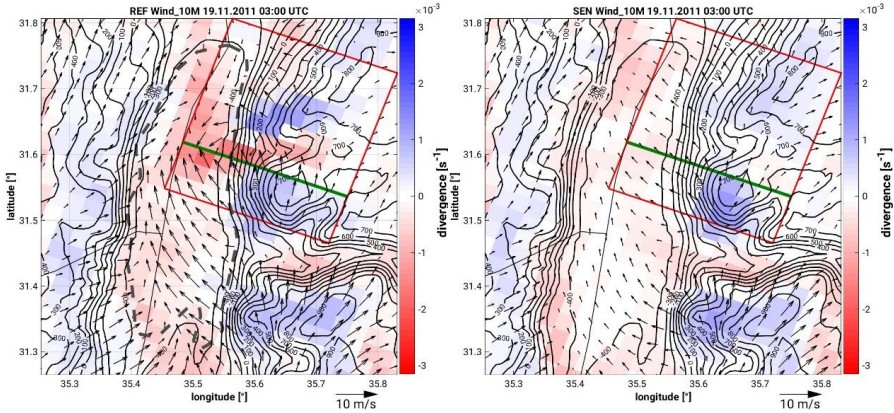



Figure 10: Spatial distribution of 10-m horizontal wind (wind vectors; m/s) and
corresponding divergence/convergence field (colour scale; s⁻¹) at 18 UTC, 20 UTC, 23
UTC on the 19 November, and 03 UTC on the 20 November 2011 from the REF[14.11] (left)
and SEN[14.11] (right) simulations. The topography is indicated by the black full isolines.
The transects (B-C-D and B'-C'-D') corresponding to the locations in which temperature
comparisons are made are indicated in Figure 10a. The green line indicates the position
of the vertical cross-section in Figure 11.






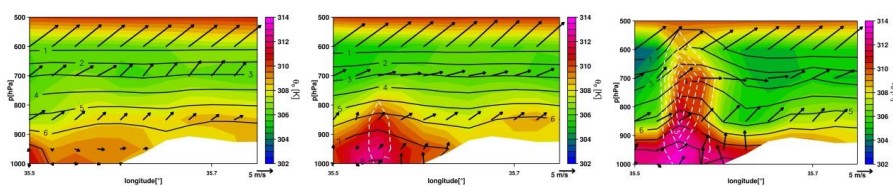



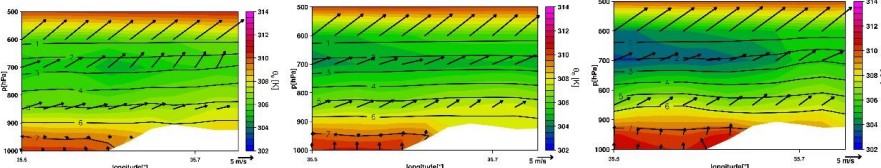




Figure 11: Vertical cross-section of equivalent potential temperature (colour scale; K),
specific humidity (black isolines; g/kg), horizontal wind vectors (north-pointing upwards,
m/s) and vertical velocity (white dashed contours with 1 m/s increments) of the REF[14.11]
(top) and SEN[14.11] (bottom) simulations at 01 UTC (left), 02 UTC (middle) and 03 UTC
(right). The location of the cross-section is indicated in Figure 10.









