# Peer review of "Near East Desertification: sensitivity of the local conditions leading to convection to the Dead Sea drying out"

_Atmospheric Chemistry and Physics, 2019_

## Referee Comment (RC1) · Anonymous Referee #1 · 27 Aug 2019

"Near East Desertification: Impact of Dead Sea drying on convective rainfall" by Khodayar and Hoerner submitted to Atmospheric Chemistry and Physics

August 27, 2019

The paper presents an interesting idea of how removing Dead sea can impact the precipitation and evaporation in the surrounding areas. Thus, it addresses the important topic of drying Dead sea in a future warmer climate and how drying of the sea alone can impact the climate. Due to its importance, the paper could be considered for the publication but some major issues have to be addressed. I list them below.

General Comments:

[Figure]

1. I do not think that you can say that you investigate the impact of drying on convective rainfall. You use only daily model output, while convective events are short-term and local events, and can thus be lost in daily output. I would thus recommend to analyze the hourly output for the entire 10-year period, and not only for two events. However, as I understand from the manuscript you have saved daily output only for that simulation so analysis of hourly events across decade-long simulations would not be possible without rerunning the simulation. I wonder if you should then at least change the title into something like "Near East Desertification: Impact of Dead Sea drying on the water cycle" or "Near East Desertification: Impact of Dead Sea drying on the water budget" or "Near East Desertification: Impact of Dead Sea drying on the local climate".

2. Why do you use high-resolution convection-permitting simulation since you analyze only daily output on decadal time-scales and daily statistics is already well represented with the coarser resolution model? The studies that you cite show that the largest benefit of using such a high resolution is at the sub-daily time scale.

3. The manuscript would further benefit from a better explanation of the domain that you simulate. I do not see how many grid points you have in x and y direction, and how large is the relaxation zone which you should take out from the analysis. How many grid points do you have at the end for the analysis? My rough estimate leads to a smaller number so the influence of the domain size has to be discussed.

4. Since you are doing the sensitivity experiment, I wonder if you really need a 10-year long period and if you could already address the problem with 1-5 years long simulations. With reducing the number of simulated years, you could use a larger domain.

5. How is the model performing over that region? You do not show any validation of the results for the reference simulation, so why should we trust the model? You even state on page 4, line 101-102 that this is the first attempt i.e., a convection-permitting model is for the first time applied in that region, so the manuscript should for sure present

some evaluation results. You already mention some papers with the observations, so maybe these can be used. Or in the absence of high-resolution observations, EOBS observations can be used as well. Of course, one should be aware of and take into account the uncertainties for different regions and fields.

6. Throughout the manuscript, you use the difference between the REF and SEN experiments, and you calculate it as REF-SEN, which is a bit strange since it is common to use the reference simulation, in your case REF, as a subtrahend. This would make a discussion and figures easier to follow.

7. I do not think that the heavy precipitation events that you analyze are well chosen. You take two events that have the same synoptic patterns, while in the introduction you mention that heavy precipitation events are associated with the three main synoptic patterns. The two chosen events are only a few days apart and their connection is not discussed. It would be more interesting to choose 1-2 events for each type and then analyze them. These would lead to more meaningful results.

8. Some plots are really difficult to read in the printed version, especially in Figure 3. In addition, not all that is shown on the plots is explained in the captions (for example in Figure 4). Please do a better caption and work on the visibility of the plots.

Specific comments:

1. Page 2, line 5-6: "Perturbation simulations..." I would call them "Sensitivity simulations..."

2. Page 2, line 13-14: You only look into the sensitivity on the presence of the lake, not really the future warmer climate. For that, you would need to modify your experiment. I would thus suggest here to explain only the influence of the lake presence, and in the final line, you can explain what would that mean for the future warmer climate.

3. Page 2, line 15-16: I do not think that you show that.

4. Page 2, line 21-23: Why on many occasions, if you find/show that for only one

event?

5. Page 3, line 39-40: A bit strange line. Please rewrite. Also, if the influence of Dead sea on local climate is already known, why do we need another study on it.

6. Page 3, line 61-65: I have the feeling that these two lines are describing the same but still say different. Please synchronize it, or if the different studies say different things, please mention it to be clearer.

7. Page 4, line 78: by "...these events..." you mean "...these heavy events..."

8. Page 4, line 88: As already mentioned above, you look into the sensitivity of climate the presence of the lake and not the climate change.

9. Page 5, line 120-122: This part needs a better explanation of the model setup. The 7km and 2.8 km domains are different (as shown in Figure 1). How many grid points do you use for each of them? How does 7 km and 2.8 km model differ in model physics? Do you use the parameterization of convection in 7 km or not?

10. Page 5, line 122-125: Here you say that you are using ECMWF IFS as a driving data for 7 km model, and later on page 6 (line 148-150) you say that the reanalysis is used. Please clarify.

11. Page 6, line 136: The more appropriate reference for the delta-two-stream approach is Ritter and Geleyn (1992). [Ritter, B., and J.-F. Geleyn, 1992. A comprehensive radiation scheme for numerical weather prediction models with potential applications in climate simulations. Mon. Wea. Rev., 120, 303–325.]

12. Page 6, line 142: Please note that the event of 14.11.2011 is not listed in Table 1.

13. Page 5-6: If you are already running the 7km simulation, maybe you should consider to use the output and compare it to 2.8 km simulation to assess the benefit of high-resolution (and or switching off convection parametrization) simulations for that region.

14. Page 6, line 153: To what soil texture do you put it? Which soil type from page 5 line 130?

15. Page 6, line 166-167: Can we talk about the trends in 10-yearlong simulations?

16. Page 7, line 173: This is how you should do the differences, but note that you do them with respect to the sensitivity simulation. See general comment 6.

17. Page 7, line 196-199: I do not understand this paragraph.

18. Page 5-8: I do not find any explanation on how do you define heavy events that you list in Table 1 or how do you classify them as localized or widespread.

19. Page 9, line 251: For consistency, please use only one name for the sensitivity experiment.

20. Page 9, line 254: I still do not understand how do you define heavy precipitation events.

21. Page 9, line 256-257: You do not show that results, but could you at least mention how much is that difference? If it is not that significant or large, I do not see why you mention in the abstract that there is that difference.

22. Page 9, line 264: I do not see reduced precipitation in the SENCLIM experiment.

23. Page 10, line 280-281: How do you define these regions? This should be explained in the methods.

24. Page 10, line 289-295: Do you always use only land points or just for Figure 3? If just for Figure 3, explain why do you do it. How is that contributing to the overall analysis?

25. Page 13, line 381-383: What is the relation between these two events? Are not they too close? Why only these two are chosen from the same period and with the same synoptic situation?

26. Page 13, line 393: Caption below Figure 7 says that this is mean precipitation and not accumulated.

27. Page 17, line 522-524: This is the third time that you mention these results, so it adds on their importance but still you do not show them in the manuscript. Either just mention it in the discussion, but if you want to discuss them in the abstract and conclusion you should consider adding these plots to the manuscript. Please note also that these differences could be larger for the hourly precipitation events i.e., more local convective events which would depend on the local evaporation sources.

---

## Referee Comment (RC2) · Anonymous Referee #2 · 28 Aug 2019

The paper "Near East Desertification: impact of Dead Sea drying on convective rainfall" by Samiro Khodayar and Johannes Hoerner presents an interesting analysis of the possible impact of the absence of the Dead Sea on the precipitation regime in the region. It is done using two simulations of 11 years (2003-2013) with the high spatial resolution (2.8 km) COSMO-CLM model, where one simulation is the reference and in the other the lake area is replaced by a soil surface. Two case studies are also considered in more details.

I have some major concerns that need to be addressed as detailed below:

Major comments:

1) Modeled mean annual precipitation: The mean annual precipitation computed by the model (Figure 5b) is quite different from observations, both in absolute magnitude and in gradients. At this region, the mean annual rain near the Mediterranean shoreline is in the range of 400-600 mm/year while over the higher topography west of the lake it can reach 500-700 mm. In the simulations presented in Figure 5b the range is from <75 mm/year at a close distance to the Mediterranean shore to 125-300 mm/year at the high topography west to the Dead Sea. The model presents much drier conditions and much larger gradients and seems not to represent well the typical more intense rain near the shore. The general effect of distance from sea on precipitation is not captured, while the orographic effect is probably well simulated. Although this paper is not focused on the effect of the Mediterranean Sea, still, as the main source of moisture to precipitation in the region, including in the study area, it is of concern that total amounts and gradients of precipitations are not represented well. The authors do not refer to this important deviation at all, not to mention explain why is it so high and why is it not harming the validity of the results and conclusions.

2) Dead Sea representation in the model: the lake form shown in Figure 1b is very noisy and different from the real lake shape and coverage. I understand this is how the lake is seen in the global data set of land use but the authors still could manually apply the actual lake shape. Furthermore, it is not stated anywhere in the paper if the salinity of the water was account for. The very high salinity reduces substantially evaporation rate compared to fresh water. Another important aspect is water temperature. What was used? This also can affect substantially evaporation and it is very different from the Mediterranean Sea temperature. All these features – lake shape, water salinity and temperature must be addressed as this is the most important feature in the simulation. The authors should note there are publications on the Dead Sea evaporation rate (e.g., Hamdani et al., 2018), so the simulated lake evaporation in the REF run can be verified.

3) Dead Sea abundance simulation: for the sensitivity analysis simulation the authors replace the lake with a soil at an elevation of 405 m below mean sea level, stating that this is the depth of the Dead Sea in the external data set, GLOBE. I find this quite strange as presently the lake level is at ~430 m below sea level; the lake's bathymetry is characterized by steep slopes and wide, flat lake floor at 720 m below mean sea level (see for example Sirota et al., 2017 among many other publications about the lake). So it is not clear what does the height of 405 m represent; if the Dead Sea will dry out, most probably the surface will be at a much lower height. Furthermore, the high gradient slopes exposed as a result of this drying can possibly affect precipitation, which is presently not considered in the paper. Also, please note, some studies

claim it will not dry out but will get to a new (possibly much lower) steady state level (Yechieli et al., 1998).

4) Dead Sea moisture transport and winds: it could be very helpful to give some background on the prevailing winds in the region and, if possible, on tracks of Dead Sea-originated moisture, possibly by backward moisture tracking analyses. For example, as the western component is mostly positive in wind direction, changes in precipitation patterns associated with Dead Sea absence are expected to be much stronger east to the lake than at its west side. This aspect is mentioned for the two case study analyzed but not in the climatologically sense.

5) Separating real effects from noise: it is hard to tell what of the effects presented in the paper are real and what are part of a noise or random error. Although the two model runs receive the exact same lateral and initial conditions, still, some differences could result from small numerical effects, not related to the Dead Sea absence. Especially, if one considers the argument in 4, above, it is not expected to have symmetrical differences on the west-east axis; however, Figure 5b (right) looks very noisy and the noise seems to have a similar pattern west and east to the lake. Could it be this noisy field of precipitation differences between the two simulations is random errors? one way to check this is to build the distribution of random differences by repeating the reference simulation few times and then consider only differences between the SEN and the REF simulations that are out of the 0.95 quantile.

Specific comments:

6) In some of the figures (e.g., Figure 2) evaporation is computed over land and lake areas and such results are hard to interpret. Obviously, the lake pixels have very high evaporation in the REF simulation and very low evaporation in SEN simulation. Could it be that this is the main control of the total volume difference between the two simulations? or, alternatively, it is just a small fraction of the total volume difference? if computation is done on land pixels only, it would be more informative in my opinion.

7) Please consider presenting the differences also in a normalized form. Are the changes described negligible or substantial? presently, it is hard to tell.

8) The authors describe in the introduction the lake level decline, which is presently > 1 m/year, but they do not state clearly that this decline is due to the massive water consumption at its upstream. One may get the impression that this substantial lake level decrease is due to climate change; this is wrong. It is possible of course that climate changes have a contribution to the lake level decrease during the last decades but it can explain much smaller decline rates comparing to the effect of water use (Lensky and Dente 2015).

9) The model spatial resolution is high, 2.8 km, and at this resolution convection can be resolved directly. However, not sure this is also true for shallow convective. Can you provide some info how was shallow convection handled? Another question is whether 2.8 km is small enough for small-scale convective typical to the Dead Sea manifested for example in the small convective rain cell size (e.g., Belachsen et al., 2017).

10) L101: Note that high resolution modelling in the region was performed by few studies, including: Hochman et al. (2018), Rostkier-Edelstein et al. (2014), Kunin et al. (2019) and possibly others.

11) L290: "…almost no difference…". I may have misunderstood the sentence, but it seems to me there are large differences in simulated evaporation in REF and SEN for A1 and A2 (Figure 3b). Also, it seems as there is more evaporation in the absence of the Dead Sea. Could it be because of the higher 2mT? Maybe there is also a change in the wind regime that could contribute to this?

12) L351: can you explain the differences in 500 hPa geopotential height?

13) L358-359: how many instances does a probability of $1^{e-6}$ represents? Could it be a single occurrence, possibly by chance?

14) L439: how MSB is differentiated from the cyclone-related wind? Does ground temperatures in this day hotter or colder than SST? Could the decreased wind near the Dead Sea be related to the higher friction caused by the change in land use? Wouldn't this differ if the ground was set to 700 m below mean sea level rather than 405 m?

15) L456: This is a good point. However, what is the temperature of the Dead Sea surface in the REF simulation? Isn't the opposite effect expected, since the Dead Sea surface temperature in November is ~25 oC (e.g., Hamdani et al., 2018)?

Minor comments

16) L183: The statement about L (from SAL) is not accurate. It measures the distance of the center of mass of precipitation from the modelled one, and the average distance of each object from the center of mass.

17) L286: north-west instead of north-east for A1

18) L307: mm per day?

19) L330: a better citation for lake evaporation would be Hamdani et al., 2018

20) L332: evaporation is probably correlated with rainfall which in turn correlated with topography. Also, soil type is often correlated with topography and rainfall.

21) L368: correct zero 0.

22) L406: gradient units should not be per km?

23) L457-458: it is hard to see the "near-surface" temperature in Figure 11, since it is plotted from 1000 hPa, while the Dead Sea is at ~1060 hPa.

24) Figure 7 caption: please check. left and right of 7a are not the REF and REF-SEN.

25) Some of the figure units should be corrected. For example, mm to mm d^-1.

References:

Belachsen, I., Marra, F., Peleg, N., Morin, E. 2017. Convective rainfall in a dry climate: relations with synoptic systems and flash-flood generation in the Dead Sea region. Hydrology and Earth System Sciences, 21, 5165-5180. doi:10.5194/hess-21-5165-2017.

Hamdani, I., Assouline, S., Tanny, J., Lensky, I.M., Gertman, I., Mor, Z. and Lensky, N.G., 2018. Seasonal and diurnal evaporation from a deep hypersaline lake: The Dead Sea as a case study. Journal of hydrology, 562, pp.155-167.

Hochman, A., Mercogliano, P., Alpert, P., Saaroni, H. and Bucchignani, E., 2018. High-resolution projection of climate change and extremity over Israel using COSMO-CLM. International Journal of Climatology, 38(14), pp.5095-5106.

Kunin, P., Alpert, P. and Rostkier-Edelstein, D., 2019. Investigation of sea-breeze/foehn in the Dead Sea valley employing high resolution WRF and observations. Atmospheric Research.

Lensky, N. and Dente, E., 2015. The hydrological proecesses driving the accelerated Dead Sea level decline in the past decades. Geological Survey of Israel Report.

Rostkier-Edelstein, D., Liu, Y., Wu, W., Kunin, P., Givati, A. and Ge, M., 2014. Towards a high-resolution climatography of seasonal precipitation over Israel. International Journal of Climatology, 34(6), pp.1964-1979.

Sirota, I., Enzel, Y. and Lensky, N.G., 2017. Temperature seasonality control on modern halite layers in the Dead Sea: In situ observations. Bulletin, 129(9-10), pp.1181-1194.

Yechieli, Y., Gavrieli, I., Berkowitz, B. and Ronen, D., 1998. Will the Dead Sea die?. Geology, 26(8), pp.755-758.

---

## Author Comment (AC1) · 11 Dec 2019

**Answers to Anonymous Reviewer #1**

*"Near East Desertification: Impact of Dead Sea drying on convective rainfall" by Khodayar and Hoerner submitted to Atmospheric Chemistry and Physics*

Dear Reviewer1:
Thanks for your comments and suggestions. We have considered all of them and improved the manuscript accordingly. In the following you can find a detail answer to all your general and specific comments.
Kind regards
Samiro Khodayar
* * *
**General Comments:**

1.   I do not think that you can say that you investigate the impact of drying on convective rainfall. You use only daily model output, while convective events are short-term and local events, and can thus be lost in daily output. I would thus recommend to analyse the hourly output for the entire 10-year period, and not only for two events. However, as I understand from the manuscript you have saved daily output only for that simulation so analysis of hourly events across decade-long simulations would not be possible without rerunning the simulation. I wonder if you should then at least change the title into something like "Near East Desertification: Impact of Dead Sea drying on the water cycle" or "Near East Desertification: Impact of Dead Sea drying on the water budget" or "Near East Desertification: Impact of Dead Sea drying on the local climate".

*We agree with the reviewer that the ideal approach would have been to analyse the hourly output for the entire 10-year period. Unfortunately, as specified in the manuscript we initially only saved daily output because of the storage capacity since our initial purpose was to assess the impact of a drying Dead sea on the climatology of the region. However, after careful inspection of our results we found interesting impacts on the precipitation field, particularly on severe events mainly of convective nature (which are rare but relevant in the area) and even more interesting results when analysing the underlying mechanisms. Even though only daily precipitation was available for the entire 10-year long simulation, the convective nature of the investigated cases was clear due to the isolated situation of the events investigated as well as their characteristics such as high local convective available potential energy. Following the need for higher temporal information new simulations with hourly outputs were performed. The approach as well as some of the results obtained is novel, therefore, we believed in the relevance of publishing this study.*
*In a follow-up publication covering a 20-year period, hourly outputs are being saved for the entire simulation. This will allow us to come back to the points raised by the reviewer. Indeed, in this follow-up publication we will investigate in more detail the impacts on the local climate. Therefore, regarding a change in the title and following the reviewer suggestion we propose the following*
*Near East Desertification: impact of Dead Sea drying on the local conditions leading to convection.*

*A comment has been included in the manuscript to clarify that the daily output supposes a limitation.*

2. Why do you use high-resolution convection-permitting simulation since you analyse only daily output on decadal time-scales and daily statistics is already well represented with the coarser resolution model? The studies that you cite show that the largest benefit of using such a high resolution is at the sub-daily time scale.

*Even when using daily outputs the use of high-resolution convection-permitting simulations is beneficial for a better representation of model characteristics and atmospheric processes leading to convective precipitation, such as topography, secondary wind circulations etc*

*Although the main benefit of high-resolution convection-permitting simulations versus parameterized convection simulations is at sub-daily time scales, particularly for summer period, as adequately pointed out by the reviewer, daily precipitation has also been seen to be affected and improved, particularly in winter time (Fosser et al. 2014).*

*Fosser, G. & Khodayar, S. & Berg, P.. (2014). Benefit of convection permitting climate model simulations in the representation of convective precipitation. Climate Dynamics. 44. 45-60. 10.1007/s00382-014-2242-1.*

*Moreover, high-resolution convection-permitting simulations on shorter time scales with hourly output are used for further investigation of underlying mechanisms leading to heavy precipitation in the area of investigation. This allows the consistency between the simulation of the events at both simulation schemes.*

*We agree with the reviewer that this is a relevant point, so we included a comment with respect to this point in the manuscript.*

3. The manuscript would further benefit from a better explanation of the domain that you simulate. I do not see how many grid points you have in x and y direction, and how large is the relaxation zone which you should take out from the analysis. How many grid points do you have at the end for the analysis? My rough estimate leads to a smaller number, so the influence of the domain size has to be discussed.

*In Figure 1a, the simulation domains at 7 km and at 2.8km km as well as the investigation domain or study area are shown, to complement this information further details such as the number of grid points in x and y direction have been included in the manuscript as suggested by the reviewer.*

*The 7 km run covers a box of 250 x 250 grid points, the 2.8 km run covers a 150 x 150 grid points box, 22500 in total, and the study area, 72 x 92 grid points, leaving between 20-40 grid points as relaxation zone in the north-south-east-west direction.*

*The influence of the domain size on the simulation and analysis has been already pointed out in literature for different regions. In this study we have discussed with experts in the region and considered past studies in the area to make sure that our larger domain is well located and large enough to have into consideration all possible relevant synoptic situations as well as the Mediterranean sea impact relevant for the development of extreme phenomena in the study area. This explanation has been included in the manuscript.*

4. Since you are doing the sensitivity experiment, I wonder if you really need a 10- year long period and if you could already address the problem with 1-5 years long simulations. With reducing the number of simulated years, you could use a larger domain.

*In our sensitivity experiment we have seen that at least 1 to 2 years spin-up time are needed. After this consideration we agree that shorter time periods could be beneficial when particular situations/periods/events have to be investigated, particularly regarding the computational time and costs of the study. After discussion with experts in the modelization of the area regarding the size of the larger domain no benefit has been found in doing so. However, the time period considered is highly beneficial for the climatic aspects considered in this analysis, which provides a novel perspective of the conditions in the area.*

5. How is the model performing over that region? You do not show any validation of the results for the reference simulation, so why should we trust the model? You even state on page 4, line 101-102 that this is the first attempt i.e., a convection-permitting model is for the first time applied in that region, so the manuscript should for sure present some evaluation results. You already mention some papers with the observations, so maybe these can be used. Or in the absence of high-resolution observations, EOBS observations can be used as well. Of course, one should be aware of and take into account the uncertainties for different regions and fields.

*In general, the observations in the area are scarce in time and space. We performed some initial comparisons with the CMORPH satellite precipitation product; however, no comparisons were included in the manuscript due to some strange values over the Dead Sea region. No validation of this satellite product has been attempted in the region to the authors knowledge.*

*Following the reviewer suggestion, we performed comparisons with EOBS data set despite the coarse resolution of the later, 0.1º, and the indication by experts in the region of the bad performance of this product in the area.*

*2.8km CTRL model_*          *EOBS_data set*
*mean over the 2004-2013 period*          *Same period*

[Figure]

*This comparison points out a general underestimation of precipitation in the north and particularly near the Mediterranean shoreline, but correctly captures the north-south gradient in the area.*
*This suggests that the model well simulates the orographic effect, while the general effect of distance from sea on precipitation is not well captured. Both the 7 km and the 2.8 km runs exhibit the same performance, thus, discarding a relationship of the biases with the grid spacing. Nevertheless, one may notice an improvement in the finer model resolution, particularly over topographic areas.*

*An additional comparison has been performed with the APHRODITE's (Asian Precipitation - Highly-Resolved Observational Data Integration Towards Evaluation) daily gridded precipitation which is the only long-term continental-scale daily product that contains a dense network of daily rain-gauge data for Asia. It has a resolution of 0.25º and is available for 1980-2007. The advantage is that it includes more rain gauge stations and it is a product widely used for validation purposes in this region of the globe. We compared the data from 2004-2007 with the respective data from the 2.8km simulation and EOBS. Please be aware of the different colormap scale between the EOBS/APHRODITE and the model simulation precipitation fields.*

*The Aphrodite data shows lower precipitation values than EOBS, but still higher than our simulation particularly close to the northern Mediterranean shoreline, over coastal-flat terrain, whereas the best agreement is again at areas dominated by complex terrain. This agrees with previous high-resolution modelling activities in the region with different models such as Rostkier-Edelstein et al. (2014) using WRF at 2 km. They suggest in this publication that inaccuracies in the gridded SST dataset used in the simulations could be responsible for the observed bias pointing out the strong sensitivity of precipitation in the Mediterranean basin to very small differences in the SST (Miglietta et al. 2011). Contrary to these results, Hochmann et al. (2018) showed with the COSMO-CLM model at 8 km resolution and driven by CMCC-CM against APHRODITE, a west-east pattern of overestimations in the coastal plains and underestimations in the mountainous regions in the seasonal precipitation, especially in the winter months (DJF).*

EOBS          APHRODITE

2.8km-CTRLsimulation-IFS forcing

[Figure]

*We performed an additional comparison for the year 2007, between the 2.8km-CTRL simulation-IFS forcing from the present study and the new simulations we are performing for the region, with hourly outputs covering the 2006-2018 period and forced by the newly available hourly ERA-5 data, also with a larger*

*simulation domain for the 2.8 km simulation. Our intention was to assess the possible impact of using different forcing data and/or domain of simulation. Our results show quite similar results between both simulations and similar biases with respect to the APHRODITA dataset, particularly at the Mediterranean shoreline, although a small improvement is shown in this area in the new model realization.*

[Figure]

*The comparison of the temporal evolution and the yearly sum for the year 2007 shows that in general the COSMO-CLM model both at 7 km and 2.8 km, quite well represents the precipitation events in the region.*

[Figure]

*We will discuss this in the paper and additionally indicate in the conclusions that further improvement/refinement in the model simulations in the region is needed despite the improvement seen by using higher resolution convection permitting model simulations.*

6. Throughout the manuscript, you use the difference between the REF and SEN experiments, and you calculate it as REF-SEN, which is a bit strange since it is common to use the reference simulation, in your case REF, as a subtrahend. This would make a discussion and figures easier to follow.

*This has been changed throughout all the manuscript. Changes include Figure 2, 3, 4 5 and 7.*

7. I do not think that the heavy precipitation events that you analyze are well chosen. You take two events that have the same synoptic patterns, while in the introduction you mention that heavy precipitation events are associated with the three main synoptic patterns. The two chosen events are only a few days apart and their connection is not discussed. It would be more interesting to choose 1-2 events for each type and then analyze them. These would lead to more meaningful results.

*We agree with the reviewer that the selection of events could attend different motivations, for example different events associated to different synoptic patterns. However, in this case it was not our motivation or purpose to investigate convective situations under different synoptic patterns, but rather cases in which the mechanisms leading to the observed differences in the precipitation field between the reference and the sensitivity experiment are noticeable different at the local-to-mesoscale. As indicated in section 3.1 and Figure 6, the selected situations where those showing a larger deviation regarding SAL (Structure-Amplitude-Location) components between both simulations. Nevertheless, we generally investigated all cases listed in table 1 to have a general idea of the mechanisms leading to the differences; however, it was out of the scope of this paper to analyse all in detail.*
*A relevant point indeed is the possible connection between both events, thanks for pointing out this. We did not discuss this in the text, but we did investigate this point during the analysis of the cases finding out that in the period in between both events no atmospheric differences could be found between the simulations. This information has been included in the text.*

8. Some plots are really difficult to read in the printed version, especially in Figure 3. In addition, not all that is shown on the plots is explained in the captions (for example in Figure 4). Please do a better caption and work on the visibility of the plots.

*We have improved the quality and size of most figures to improve their visibility. Additionally, we have carefully examined all captions to include relevant information that was missing.*

***Specific comments:***

1.  Page 2, line 5-6: "Perturbation simulations. . ." I would call them "Sensitivity simulations. . ."

*Changed*

2. Page 2, line 13-14: You only look into the sensitivity on the presence of the lake, not really the future warmer climate. For that, you would need to modify your experiment.

I would thus suggest here to explain only the influence of the lake presence, and in the final line, you can explain what would that mean for the future warmer climate.
*We agree with the reviewer that we are not simulating the future, however, literature agrees that the future climate in the region includes a drier Dead Sea. In this sense, the expected ongoing lake level is expected to have the described consequences on the local climate. To clarify this point and avoid any misunderstanding we rewrote the paragraph following the reviewer's suggestion.*

3. Page 2, line 15-16: I do not think that you show that.

*This is not explicitly shown in the manuscript, mainly because the differences are not significant. Therefore, following the reviewer's suggestion regarding this point we removed this information from the abstract.*

4. Page 2, line 21-23: Why on many occasions, if you find/show that for only one event?

*Even though we just show as example a more detailed analysis of two selected cases, we did investigate for all other events listed in Table 1, which were the main mechanisms leading to the differences between both simulations. This was also to evidence that the cases examined were not unique. This information has been included in the manuscript.*

5. Page 3, line 39-40: A bit strange line. Please rewrite. Also, if the influence of Dead sea on local climate is already known, why do we need another study on it.

*The paragraph has been rewritten.*
*Even though the influence of the Dead Sea on local climate has been evidenced in several publications starting in 1939, Ashbel et al., the advances in the last decade regarding observational and computational capacities allow us to better understand the consequences of the sea level decline, which is furthermore a continuous process rather than a static change.*

6. Page 3, line 61-65: I have the feeling that these two lines are describing the same but still say different. Please synchronize it, or if the different studies say different things, please mention it to be clearer.

*Modified*

7. Page 4, line 78: by ". . .these events. . ." you mean ". . .these heavy events. . ."

*Included*

8. Page 4, line 88: As already mentioned above, you look into the sensitivity of climate the presence of the lake and not the climate change.

*Modified*

9. Page 5, line 120-122: This part needs a better explanation of the model setup. The 7km and 2.8 km domains are different (as shown in Figure 1). How many grid points do you use for each of them? How does 7 km and 2.8 km model differ in model physics? Do you use the parameterization of convection in 7 km or not?

*A paragraph is included extending the information regarding the 7 and 2.8 km runs.*

10. Page 5, line 122-125: Here you say that you are using ECMWF IFS as a driving data for 7 km model, and later on page 6 (line 148-150) you say that the reanalysis is used. Please clarify.

*This has been corrected.*

11. Page 6, line 136: The more appropriate reference for the delta-two-stream approach is Ritter and Geleyn (1992). [Ritter, B., and J.-F. Geleyn, 1992. A comprehensive radiation scheme for numerical weather prediction models with potential applications in climate simulations. Mon. Wea. Rev., 120, 303–325.]

*We agree with the reviewer, thanks for pointing out this.*

12. Page 6, line 142: Please note that the event of 14.11.2011 is not listed in Table 1.

*Thanks for noticing this has been correctly indicated.*

13. Page 5-6: If you are already running the 7km simulation, maybe you should consider to use the output and compare it to 2.8 km simulation to assess the benefit of high-resolution (and or switching off convection parametrization) simulations for that region.

*The 7 km simulation has been run only in reference mode (CTRL). The benefit of the CCLM-2.8 km high-resolution convection permitting simulations versus CCLM-7 km with parameterized convection has been already investigated in detail in the past e.g. in Fosser et al. (2014). We additionally performed some comparisons, please see answer to comment number 5.*

14. Page 6, line 153: To what soil texture do you put it? Which soil type from page 5 line 130?

*The soil types are histosols, clay and loamy clay, visible in Figure 1 as they are bordering the Dead Sea.*

15. Page 6, line 166-167: Can we talk about the trends in 10-yearlong simulations?

*Modified*

16. Page 7, line 173: This is how you should do the differences, but note that you do them with respect to the sensitivity simulation. See general comment 6.

*Thank you for pointing out this, as previously explained we modified all calculations in the manuscript.*

17. Page 7, line 196-199: I do not understand this paragraph.

*This corresponds to the classification in Table 1, which has been clarified now in the text.*

18. Page 5-8: I do not find any explanation on how do you define heavy events that you list in Table 1 or how do you classify them as localized or widespread.

*The events were selected with an area mean difference in precipitation larger than 0.1 mm/d. localised or widespread nature of the vents was assessed visually for each case. Due to the low temporal resolution of the simulation it was not possible to use any predefined index such as tau, which requires more frequent outputs.*

19. Page 9, line 251: For consistency, please use only one name for the sensitivity experiment.

*Changed*

20. Page 9, line 254: I still do not understand how do you define heavy precipitation events.

*Please see 18*

21. Page 9, line 256-257: You do not show that results, but could you at least mention how much is that difference? If it is not that significant or large, I do not see why you mention in the abstract that there is that difference.

*Initially we included a table specifying the number of dry and wet days as well as the differences between simulations. However, since the differences where almost negligible, just a few days as described in the text, we considered this table was not relevant enough to be included in the final version. We agree with the reviewer that the differences are not significant, therefore we removed this information from the abstract.*

22. Page 9, line 264: I do not see reduced precipitation in the SENCLIM experiment.

*For each event the percentage of precipitation change has been calculated and included in the table. Additionally, the total percentage of change for the whole period, which corresponds to a reduction of about 0.5 % in the SEN simulation has been calculated and all this information has been included in the manuscript.*

23. Page 10, line 280-281: How do you define these regions? This should be explained in the methods.

*This information has been included in the methodology as suggested by the reviewer.*

24. Page 10, line 289-295: Do you always use only land points or just for Figure 3? If just for Figure 3, explain why do you do it. How is that contributing to the overall analysis?

- *Figure 2: all GP for all calculations are considered. In the following the example of evaporation including all GP(left) and only land points (right) is shown.*

[Figure]

- *Figure 3: Only land points for evap and all GP for the rest*
  *The Dead Sea grids have very high evaporation in the REF simulation and very low evaporation in SEN simulation, this difficult the interpretation of the results, thus we removed this effect in Figure 3 for evaporation, also because of the separation in 4 subdomains. In the following the example of evaporation including all GP(left) and only land points (right) for Area 1 is shown.*

[Figure]

25. Page 13, line 381-383: What is the relation between these two events? Are not they too close? Why only these two are chosen from the same period and with the same synoptic situation?

*These two events where those showing the larger difference between the REF and SEN simulations (FIG SAL), the synop situation and the fact that they were close in time were no relevant factors for our analysis*

*rather the mechanisms responsible for the differences observed between both simulations. Even though the two cases were close in time and a connection was to be expected between the first and the second periods, after analysing atmospheric conditions between these two periods similar atmospheric conditions were found.*

26. Page 13, line 393: Caption below Figure 7 says that this is mean precipitation and not accumulated.

*This has been corrected*

27. Page 17, line 522-524: This is the third time that you mention these results, so it adds on their importance but still you do not show them in the manuscript. Either just mention it in the discussion, but if you want to discuss them in the abstract and conclusion you should consider adding these plots to the manuscript. Please note also that these differences could be larger for the hourly precipitation events i.e., more local convective events which would depend on the local evaporation sources.

*As previously discussed, we agree with the reviewer, therefore we removed this in the abstract, and in the conclusion just mentioned that number of dry/wet days is not largely affected, suggesting that these differences could be larger for hourly precipitation events to point out the limitation in this study and a possible future aspect to be studied.*

---

## Author Comment (AC2) · 11 Dec 2019

**Answers to Anonymous Reviewer #2**

*"Near East Desertification: Impact of Dead Sea drying on convective rainfall" by Khodayar and Hoerner submitted to Atmospheric Chemistry and Physics*

Dear Reviewer2:

We have corrected this manuscript following all your comments and suggestions. In the following you can find a detail answer to all your general and specific comments.

Kind regards

Samiro Khodayar
* * *
**General Comments:**

Major comments:

1) Modeled mean annual precipitation: The mean annual precipitation computed by the model (Figure 5b) is quite different from observations, both in absolute magnitude and in gradients. At this region, the mean annual rain near the Mediterranean shoreline is in the range of 400-600 mm/year while over the higher topography west of the lake it can reach 500-700 mm. In the simulations presented in Figure 5b the range is from <75 mm/year at a close distance to the Mediterranean shore to 125-300 mm/year at the high topography west to the Dead Sea. The model presents much drier conditions and much larger gradients and seems not to represent well the typical more intense rain near the shore. The general effect of distance from sea on precipitation is not captured, while the orographic effect is probably well simulated. Although this paper is not focused on the effect of the Mediterranean Sea, still, as the main source of moisture to precipitation in the region, including in the study area, it is of concern that total amounts and gradients of precipitations are not represented well. The authors do not refer to this important deviation at all, not to mention explain why is it so high and why is it not harming the validity of the results and conclusions.

*We have performed an extensive analysis/validation of the precipitation field using EOBS and APHRODITE information, the results show in agreement with other modelling efforts in the region a general underestimation with particular focus on the near-coast flat areas and better results over the complex areas. We have demonstrated that neither the simulations domain, the forcing data or the grid spacing used are the main reason for this bias. Indeed, closer results are obtained for the finer resolution simulations. Revising past simulation and validation exercises in the region in past publications we found out that similar biases have been identified in the past, also with different model schemes. In these publications it is pointed out inaccuracies in the SST as the reason for the biases in the precipitation field in the Mediterranean coastal area. Generally, relevant inaccuracies are identified in the SST field forcing our simulations, which have been demonstrated in the past to have a significant impact in the simulation of precipitation in the region. It is out of the scope of this paper to demonstrate this relevance, but we do agree with the reviewer that this is a relevant point to be discussed and to be investigated. Therefore, we*

*have included this discussion in the manuscript and proposed in the conclusions the need to investigate, for example though sensitivity experiments the relevance of SST to obtain a more accurate precipitation field.*

2) Dead Sea representation in the model: the lake form shown in Figure 1b is very noisy and different from the real lake shape and coverage. I understand this is how the lake is seen in the global data set of land use but the authors still could manually apply the actual lake shape. Furthermore, it is not stated anywhere in the paper if the salinity of the water was account for. The very high salinity reduces substantially evaporation rate compared to fresh water. Another important aspect is water temperature. What was used? This also can affect substantially evaporation and it is very different from the Mediterranean Sea temperature. All these features – lake shape, water salinity and temperature must be addressed as this is the most important feature in the simulation. The authors should note there are publications on the Dead Sea evaporation rate (e.g., Hamdani et al., 2018), so the simulated lake evaporation in the REF run can be verified.

*We agree with the reviewer that differences exist between the reality and the modelled Dead Sea, whose characteristics are given by the global data set. It was out of the scope of this article to improve this representation in our model simulations, but we agree with the reviewer that this is a relevant point. Therefore, we will investigate this point and the sensitivity of our simulations to this in the new simulations we are performing in the area as indicated in the conclusions. The salinity is not considered in the Dead Sea simulations. Nevertheless, in the PhD thesis of Jutta Metzger/Vüllers, and the corresponding paper,*

*Wind Systems and Energy Balance in the Dead Sea Valley, 2017 | dissertation-thesis, DOI: 10.5445/KSP/1000072084*

*Dead Sea evaporation by eddy covariance measurements vs. aerodynamic, energy budget, Priestley–Taylor, and Penman estimates. Hydrology and Earth System Sciences. 2018-02-09 | journal-article, DOI: 10.5194/hess-22-1135-2018*

*it has been demonstrated that wind speed and vapour pressure deficit (humidity) are governing factors for evaporation in the Dead Sea valley, being the influence of salinity low as assessed by measurements and calculations. During the measurement campaign of the DESERVE project evaporation measurements were performed for a period of one year, which have been used in this publication as reference. We have included in the conclusions some sentences pointing out these issues raised by the reviewer, which we agree have to be considered but were out of the scope of this study.*

3) Dead Sea abundance simulation: for the sensitivity analysis simulations the authors replace the lake with a soil at an elevation of 405 m below mean sea level, stating that this is the depth of the Dead Sea in the external data set, GLOBE. I find this quite strange as presently the lake level is at ~430 m below sea level; the lake's bathymetry is characterized by steep slopes and wide, flat lake floor at 720 m below mean sea level (see for example Sirota et al., 2017 among many other publications about the lake). So it is not clear what does the height of 405 m represent; if the Dead Sea will dry out, most probably the surface will be at a much lower height. Furthermore, the high gradient slopes exposed as a result of this drying can possibly affect precipitation, which is presently not considered in the paper. Also, please note, some studies claim it will not dry out but will get to a new (possibly much lower) steady state level (Yechieli et al., 1998).

*The remaining flat floor of the lake at some level above 720 m will be much smaller than the actual lake area. The dry level will be higher than 720 m because of the huge amount of NaCl in the valley. We agree with the reviewer that there is discussion about the possibility of the lake never drying out, they indeed point out that a wet swamp of semi-crystalline salt would remain, even if there is no more inflow of fresh water in the valley. It was out of the scope of this paper to explore or discuss this or further possibilities. However, we agree with the reviewer about the relevance of this point, for that reason in the new set of simulations*

*we are performing (follow-up article) we are considering "intermediate condition/situation of the Dead sea" in addition to the more extreme condition, totally dry, investigated in this publication.*

4) Dead Sea moisture transport and winds: it could be very helpful to give some background on the prevailing winds in the region and, if possible, on tracks of Dead Sea-originated moisture, possibly by backward moisture tracking analyses. For example, as the western component is mostly positive in wind direction, changes in precipitation patterns associated with Dead Sea absence are expected to be much stronger east to the lake than at its west side. This aspect is mentioned for the two case study analyzed but not in the climatologically sense.

*The article from Metzger et al. (2017) investigates in detail the climatology of the winds in the region. This information and corresponding reference is included in the article. It is not possible to recalculate backward moisture trajectories over the simulations performed given the resolution of our output and the impossibility of reproducing the simulations. External Lagrangian schemes could be used such as the freely available HYSPLIT software; however, this uses different model information that the one in this publication and validation will be needed to demonstrate the consistency of the results. We intend to include these calculations in the follow-up simulations we are performing in the region to complement this information.*

5) Separating real effects from noise: it is hard to tell what of the effects presented in the paper are real and what are part of a noise or random error. Although the two model runs receive the exact same lateral and initial conditions, still, some differences could result from small numerical effects, not related to the Dead Sea absence. Especially, if one considers the argument in 4, above, it is not expected to have symmetrical differences on the west-east axis; however, Figure 5b (right) looks very noisy and the noise seems to have a similar pattern west and east to the lake. Could it be this noisy field of precipitation differences between the two simulations is random errors? one way to check this is to build the distribution of random differences by repeating the reference simulation few times and then consider only differences between the SEN and the REF simulations that are out of the 0.95 quantile.

*Unfortunately, it is not possible to repeat the reference simulation since the computation system in which we run this simulations is not available anymore. However, we can confirm that at the moment of realization of these simulations we did run the same simulation in two different machines and we obtained the same results. Also the fact that we observe the same results in the 10-year long simulations and in the events simulated for several days, furthermore in many different events confirms that the effects presented in this paper are not random errors or noise.*

**Specific comments:**

6) In some of the figures (e.g., Figure 2) evaporation is computed over land and lake areas and such results are hard to interpret. Obviously, the lake pixels have very high evaporation in the REF simulation and very low evaporation in SEN simulation. Could it be that this is the main control of the total volume difference between the two simulations? or, alternatively, it is just a small fraction of the total volume difference? if computation is done on land pixels only, it would be more informative in my opinion.

*Evaporation is only computed over land in Figure 3 to facilitate the interpretation of the results.*

*Figure 2: all GP for all calculations are considered. In the following the example of evaporation including all GP(left) and only land points (right) is shown.*

[Figure]

*Figure 3: Only land points for evap and all GP for the rest*

*The Dead Sea grids have very high evaporation in the REF simulation and very low evaporation in SEN simulation, this difficult the interpretation of the results, thus we removed this effect in Figure 3 for evaporation, also because of the separation in 4 subdomains. In the following the example of evaporation including all GP(left) and only land points (right) for Area 1 is shown.*

[Figure]

7) The authors describe in the introduction the lake level decline, which is presently > 1 m/year, but they do not state clearly that this decline is due to the massive water consumption at its upstream. One may get the impression that this substantial lake level decrease is due to climate change; this is wrong. It is possible of course that climate changes have a contribution to the lake level decrease during the last decades but it can explain much smaller decline rates comparing to the effect of water use (Lensky and Dente 2015).

*This information has been included in the manuscript as we agree with the reviewer this is relevant for the readers.*

8) The model spatial resolution is high, 2.8 km, and at this resolution convection can be resolved directly. However, not sure this is also true for shallow convective. Can you provide some info how was shallow convection handled? Another question is whether 2.8 km is small enough for small-scale convective typical to the Dead Sea manifested for example in the small convective rain cell size (e.g., Belachsen et al., 2017).

*This information has been included in the description of the model.*

9) L101: Note that high resolution modelling in the region was performed by few studies, including: Hochman et al. (2018), Rostkier-Edelstein et al. (2014), Kunin et al. (2019) and possibly others.

*This information has been included in the text.*

10) L290: "…almost no difference…". I may have misunderstood the sentence, but it seems to me there are large differences in simulated evaporation in REF and SEN for A1 and A2 (Figure 3b). Also, it seems as there is more evaporation in the absence of the Dead Sea. Could it be because of the higher 2mT? Maybe there is also a change in the wind regime that could contribute to this?

*Evaporation difference is negligible in the May to November period.*

11) L351: can you explain the differences in 500 hPa geopotential height?

*Differences in the near-surface conditions impact surface pressure as well as wind circulations, this information is transported upwards in the atmosphere locally and remotely, which results in the weak changes in the upper-atmospheric levels in this case the 500 hPa geopotential height.*

12) L358-359: how many instances does a probability of 1^e-6 represents? Could it be a single occurrence, possibly by chance?

*A probability of 1^e-6 represents approximately 22 instances. The total number of instances is 21759840, so one instance would be represented with a probability of approximately 5^e-8.*

13) L439: how MSB is differentiated from the cyclone-related wind? Does ground temperatures in this day hotter or colder than SST? Could the decreased wind near the Dead Sea be related to the higher friction caused by the change in land use? Wouldn't this differ if the ground was set to 700 m below mean sea level rather than 405 m?

*The timing, characteristics and evolution of the horizontal and vertical air flow helped us identify the MSB. Ground temperature in this day varies between 18° and 31°, whereas temperature over the Dead Sea varies only slightly between 25-26°.The temperature difference between the cooler maritime air mass and the warmer valley in REF result in the downward penetration of the MSB.*

*To give an accurate answer to the last two questions we believe it would be necessary to perform some sensitivity experiments to demonstrate these hypothesis.  However, we believe that the change in land use is a contributing factor to the decrease wind, but not the only one, and the depth of the Dead Sea in the SEN simulation would not have changed the observed dynamics. It would have rather  enhanced the behaviour to more marked temperature differences.*

14) L456: This is a good point. However, what is the temperature of the Dead Sea surface in the REF simulation? Isn't the opposite effect expected, since the Dead Sea surface temperature in November is ~25 oC (e.g., Hamdani et al., 2018)?

*The ground temperature in SEN is higher than the surface temperature in REF between 8UTC and 13UTC. At point B, the mean surface temperature on the 18.11.2011 in REF is about 26°, whereas the mean ground temperature in SEN is about 19°.*

**Minor comments**

15) L183: The statement about L (from SAL) is not accurate. It measures the distance of the center of mass of precipitation from the modelled one, and the average distance of each object from the center of mass.

*Corrected*

16) L286: north-west instead of north-east for A1

*Corrected*

17) L307: mm per day?

*mm per month, as indicated in the caption of figure 4 monthly mean values calculated using daily mean values are presented. This has been corrected.*

18) L330: a better citation for lake evaporation would be Hamdani et al., 2018

*This reference has been included*

19) L332: evaporation is probably correlated with rainfall which in turn correlated with topography. Also, soil type is often correlated with topography and rainfall.

*Yes, we agree.*

20) L368: correct zero 0.

*Corrected*

21) L406: gradient units should not be per km?

*Corrected*

22) L457-458: it is hard to see the "near-surface" temperature in Figure 11, since it is plotted from 1000 hPa, while the Dead Sea is at ~1060 hPa.

*Unfortunately, this is the lowest level available as pressure level.*

23) Figure 7 caption: please check. left and right of 7a are not the REF and REF-SEN.

*Corrected*

24) *Some of the figure units should be corrected. For example, mm to mm d^-1.*

This has been corrected.

*References:*

*Belachsen, I., Marra, F., Peleg, N., Morin, E. 2017. Convective rainfall in a dry climate: relations with synoptic systems and flash-flood generation in the Dead Sea region. Hydrology and Earth System Sciences, 21, 5165-5180. doi:10.5194/hess-21-5165-2017.*

*Hamdani, I., Assouline, S., Tanny, J., Lensky, I.M., Gertman, I., Mor, Z. and Lensky, N.G., 2018. Seasonal and diurnal evaporation from a deep hypersaline lake: The Dead Sea as a case study. Journal of hydrology, 562, pp.155-167.*

*Hochman, A., Mercogliano, P., Alpert, P., Saaroni, H. and Bucchignani, E., 2018. High-resolution projection of climate change and extremity over Israel using COSMO-CLM. International Journal of Climatology, 38(14), pp.5095-5106.*

*Kunin, P., Alpert, P. and Rostkier-Edelstein, D., 2019. Investigation of sea-breeze/foehn in the Dead Sea valley employing high resolution WRF and observations. Atmospheric Research.*

*Lensky, N. and Dente, E., 2015. The hydrological proecesses driving the accelerated Dead Sea level decline in the past decades. Geological Survey of Israel Report.*

*Rostkier-Edelstein, D., Liu, Y., Wu, W., Kunin, P., Givati, A. and Ge, M., 2014. Towards a high-resolution climatography of seasonal precipitation over Israel. International Journal of Climatology, 34(6), pp.1964-1979.*

*Sirota, I., Enzel, Y. and Lensky, N.G., 2017. Temperature seasonality control on modern halite layers in the Dead Sea: In situ observations. Bulletin, 129(9-10), pp.1181-1194.*

*Yechieli, Y., Gavrieli, I., Berkowitz, B. and Ronen, D., 1998. Will the Dead Sea die?. Geology, 26(8), pp.755-758.*

---

## Referee Report (RR1)

I have read the author response to the comments. I understand there are limitations related to possibility to repeat simulations and availability of simulation data. But in the present state, the main message of this work, in my opinion, is basing on simulations that are not representing reality in few ways, that are already mentioned in my previous major comments (4 out of 5) and are summarized below:

1) Precipitation totals are substantially underestimated by the REF simulation, especially on the shore but also in other regions. The authors show the same situation also when comparing to other datasets. They can further check it using stations from the Israeli Meteorological Service (IMS) which is freely available and includes many stations. Please see below the 1981-2010 mean annual rainfall from IMS (right) compared to the REF climatological run (left). My worry is that this underestimation may indicate a problem with the main moisture source for the precipitation in the Dead Sea.

I am not clear how this discrepancy is handled. The authors refer to Rostkier-Edelstein 2014 paper, but in this work the fit with observations is much better and at the coast there is actually an overestimation of precipitation rather than underestimation.

[Figure]

2) Lake evaporation: If I understood correctly, the lake evaporation is handled as regular sea water. But the Dead Sea is much more saline than sea water! Therefore, evaporation at the REF run should be lower than the simulated for regular sea.

The authors refer to Metzger et al. 2017 claiming that vapor pressure deficit being an important factor, rather than salinity, but exactly here salinity is considered, because saturated vapor pressure near the water is multiplied by the water activity that is reduced with salinity, and thus affecting the deficit.

A good way to check the reference run would be to compare the simulated the annual lake evaporation with values published by Hamdani et al. (2018) [about 1130 mm/year for 2016-2017]. It seems that lake evaporation is computed, and is painted in magenta in Figure 5a, but this presents a range of 500-2000 mm/year, so it is hard to tell. I suggest to provide the annual lake evaporation and check how does it fit with observations. If it well fits, this is a very good indication for the model ability to represent this process, but otherwise, it would be a serious problem for the main claim of this study.

3) The elevation of 405 and missing exposed steep slopes of the empty lake: I could not understand the authors reply. Yes, the lake area would be a bit smaller when is filled by soil, but not a lot, since the lake bottom is wide and the lake slopes are steep (e.g., Sirota et al., 2017). Yes, the bottom would not be at 720 mbsl (I did not claim it will) due to precipitation of NaCl, but surely not at 405 mbsl, which is already 25 m higher than present day lake level. I did not find an answer to why an elevation of 405 mbsl was selected and what about the exposed steep slopes and their potential effect on precipitation generation.

4) Separating real effects from noise: the authors write in response to this comment that they obtained the same result in two different machines and that they "observe the same results in the 10-year long simulations and in the events simulated for several days, furthermore in many different events confirms that the effects presented in this paper are not random errors or noise". I would appreciate showing this in more details. What do you mean – the same result in 10-years? do you get this pattern for each year separately?

Also, the authors did not answer why we do not see the expected larger effect on the eastern side comparing to the western side.

In summary – based on my understanding – at its present state, the paper shows:

**Comparison of modeled precipitation in the Dead Sea region under two different hypothetical land use scenarios: 1) The Dead Sea is a lake with regular sea water, 2) The lake area is filled by soil to a level of 405 mbsl**

Unless it is proved otherwise: scenario 1 is not representing the present conditions (points 1 and 2 above) and scenario 2 is not representing Dead Sea drying (point 3 above).

If this is the case, it is ok to leave the results as they are BUT the title, the "story" told in this paper, the objectives and the conclusions must be adjusted.

---

## Author Response (AR2)

**Answers to Anonymous Reviewer #1**

*"Near East Desertification: Impact of Dead Sea drying on convective rainfall" by Khodayar and Hoerner submitted to Atmospheric Chemistry and Physics*

Dear Reviewer1:
Thanks for your comments and suggestions. We understand your point and we agree with it. We have modified the manuscript to better reflect the "story line". In the following you can find a more detail answer to your comments (in blue).
Kind regards
Samiro Khodayar
* * *
I have read the author response to the comments. I understand there are limitations related to possibility to repeat simulations and availability of simulation data. But in the present state, the main message of this work, in my opinion, is basing on simulations that are not representing reality in few ways, that are already mentioned in my previous major comments (4 out of 5) and are summarized below (points 1 to 4 below).

*We understand and agree with the point of view and comments of the reviewer. We believe incorporating this point of view in the "story line" of the manuscript will improve it and make it clearer to the reader. Therefore, we have tried to better explain and clarify in every relevant part of the manuscript the fact that we are assuming certain aspects of the modelling exercise, which do not represent accurately reality. This is due to model and external input data limitations.*

1) Precipitation totals are substantially underestimated by the REF simulation, especially on the shore but also in other regions. The authors show the same situation also when comparing to other datasets. They can further check it using stations from the Israeli Meteorological Service (IMS) which is freely available and includes many stations. Please see below the 1981-2010 mean annual rainfall from IMS (right) compared to the REF climatological run (left). My worry is that this underestimation may indicate a problem with the main moisture source for the precipitation in the Dead Sea. I am not clear how this discrepancy is handled. The authors refer to Rostkier-Edelstein 2014 paper, but in this work the fit with observations is much better and at the coast there is actually an overestimation of precipitation rather than underestimation.

In the first review, the reviewer1 and 2 asked us about these discrepancies in the precipitation field. As a consequence of this, we performed an intensive validation exercise with all adequate available data sets we could find at the time, EOBS suggested by the reviewer as well as *APHRODITE* data set suggested by experts in the area. The results show a general underestimation with particular focus on the near-coast flat areas and better results over the complex areas. We additionally performed new simulations demonstrating that neither the simulation domain, the forcing data or the grid spacing used are the main reason for this bias. Indeed, closer results were obtained for the finer resolution simulations.

We additionally focused on particular periods, example 2007, common to all data sets and new simulations, and demonstrated that despite discrepancies in the mean spatial precipitation field, the comparison of the temporal evolution and the yearly sum for the particular periods quite well represented the precipitation events in the region.

[Figure]

In addition to that, the revision of past modelling investigations / literature in the region showed different biases in different models and resolutions. Some pointed out inaccuracies in the SST as the reason for the biases in the precipitation field in the Mediterranean coastal area, which is out of the scope of this paper to be demonstrated for our particular simulation.

We agree with the reviewer that these biases could affect the moisture sources for precipitation in the Dead Sea, however there are two relevant aspects to point out, a)  The same bias is present in the ref and in the sen simulations; therefore, the bases for studying the sensitivity of HP in the region to modified conditions in the Dead sea (drying out the water) will be the same.

b)  The dominant wind conditions in the period and for the investigated cases are from the south, which reduces the possible impact of a dry bias on the north-western Mediterranean shore.

Nevertheless, we do agree with the reviewer that this is a relevant point to be discussed and further investigated. We have been able to identify several modelling aspects, which are not responsible of the biases present in the simulations of precipitation, namely, domain size, forcing data, grid spacing. We have included in the manuscript a discussion about the need to further investigate and correct this biases pointing out the possible impact on HP modelling in the region.

2) Lake evaporation: If I understood correctly, the lake evaporation is handled as regular sea water. But the Dead Sea is much more saline than sea water! Therefore, evaporation at the REF run should be lower than the simulated for regular sea.

The authors refer to Metzger et al. 2017 claiming that vapor pressure deficit being an important factor, rather than salinity, but exactly here salinity is considered, because saturated vapor pressure near the water is multiplied by the water activity that is reduced with salinity, and thus affecting the deficit. A good way to check the reference run would be to compare the simulated the annual lake evaporation with values published by Hamdani et al. (2018) [about 1130 mm/year for 2016-2017]. It seems that lake evaporation is computed, and is painted in magenta in Figure 5a, but this presents a range of 500-2000 mm/year, so it is hard to tell. I suggest to provide the annual lake evaporation and check how does it fit with observations. If it well fits, this is a very good indication for the model ability to represent this process, but otherwise, it would be a serious problem for the main claim of this study.

Yes, the model handles Dead Sea water as sea water. We already included in the previous version a comparison with observations as suggested by the reviewers

"Over the Dead Sea, the simulated average annual evaporation for the period under consideration is in the order of 1500-1800 mm/y, in contrast to the values in the deserts east and south, where the evaporation is less than 20 mm/y. Observed annual evaporation of this lake is known to be about 1500 mm, about 1130 mm/year for 2016-2017, and to vary with the salinity at the surface of the lake and freshening by the water inflow (Dayan and Morin 2006; Hamdani et al. 2018)."

3) The elevation of 405 and missing exposed steep slopes of the empty lake: I could not understand the authors reply. Yes, the lake area would be a bit smaller when is filled by soil, but not a lot, since the lake bottom is wide and the lake slopes are steep (e.g., Sirota et al., 2017). Yes, the bottom would not be at 720 mbsl (I did not claim it will) due to precipitation of NaCl, but surely not at 405 mbsl, which is already 25 m higher than present day lake level. I did not find an answer to why an elevation of 405 mbsl was selected and what about the exposed steep slopes and their potential effect on precipitation generation.

Again here we are limited by the ability of the model and the external data set to represent the reality of the region, and clearly this is the case. As described in the text, the shape, depth, soil types and characteristics of the Dead Sea itself and the region in general are imposed by the choice of external data sets, such as GLOBE from NOAA and HWSD from FAO in addition to the model grid spacing. We agree with the reviewer that a more realistic description of the area and conditions would be more advisable, we are trying to improve this point in our future simulations for the area, but it will also require future efforts in this direction.

4) Separating real effects from noise: the authors write in response to this comment that they obtained the same result in two different machines and that they "observe the same results in the 10-year long simulations and in the events simulated for several days, furthermore in many different events confirms that the effects presented in this paper are not random errors or noise". I would appreciate showing this in more details. What do you mean – the same result in 10-years? do you get this pattern for each year separately? Also, the authors did not answer why we do not see the expected larger effect on the eastern side comparing to the western side.

We had two performed twice the same simulation in two different machines due to problems with the original machine in which the simulations were performed. To prove that not large differences were present in the different simulations due to the machine used we performed comparisons of some of the model outputs, e.g. the precipitation field. We did not find any significant deviation in our simulations. Unfortunately, because of storage capacity limitations we deleted the "old" simulations and the comparisons performed.

Regarding the sentence "*the fact that we observe the same results in the 10-year long simulations and in the events simulated for several days… confirms that the effects presented in this paper are not random errors or noise.* ", means that the same HPEs are simulated when performing the NWP simulations for short periods of days, than in the 10-year long simulations.

We could not give a definitive answer based on the information we have concerning the symmetrical differences on the west-east axis in Figure 5b (right), however, we believe the precipitation field itself shows also a symmetrical distribution, which has clearly an impact on the difference field, and the differences we are referring to are less than about 2-3 mm/y.

In summary – based on my understanding – at its present state, the paper shows:
**Comparison of modeled precipitation in the Dead Sea region under two different hypothetical land use scenarios: 1) The Dead Sea is a lake with regular sea water, 2) The lake area is filled by soil to a level of 405 mbsl**
Unless it is proved otherwise: scenario 1 is not representing the present conditions (points 1 and 2 above) and scenario 2 is not representing Dead Sea drying (point 3 above).
If this is the case, it is ok to leave the results as they are BUT the title, the "story" told in this paper, the objectives and the conclusions must be adjusted.

As mentioned before, we agree with the reviewer that certain assumptions have been made, which do not represent accurately the present or future reality. Therefore, we have accordingly modified the manuscript to clarify this point and the purpose of the manuscript to the reader.

We believe in most modelling realizations the limitations given and previously discussed will make very difficult to accurately represent the true conditions, although improvements could be applied. Moreover, although the topography and depth of the Dead Sea are not well represented in the model scenario, we have demonstrated that local conditions are sensible to the drying out of the lake. Therefore, we propose the following change in the title of the manuscript:

*"Near East Desertification: sensitivity of the local conditions leading to convection to the Dead Sea drying out"*

The objective of the paper has been defined as follows: *"the sensitivity of the local conditions to the drying out of the Sea is investigated focusing on the conditions leading to heavy precipitating convection in the region."*

We included in the methodology and few sentences regarding the limitations of the model and the biases found for the reader to be aware of this issue *"We have to point out that the external data sets commonly employed describing relevant features of the Dead Sea region, such as the depth, shape and orography of the Dead Sea, as well as water characteristics at the reference run, do not accurately represent the reality. Also biases in relation to the precipitation field and evaporation over the Dead Sea have to be considered."*

Also we reflected in the conclusions this issue, for example in: "*Furthermore, a more realistic representation of the lake shape, water salinity and temperature, as well as Dead Sea abundance and depth must be addressed to more accurately describe present and expected future conditions. In the present study, limitations found in this direction in relation to model and external data set descriptions, as well as identified biases regarding for example moisture sources for HP in the region, MSB and Dead Sea evaporation, are expected to impact our results, and have to be improved in future efforts in the region. In a further step, the authors will investigate some of these issues in more detail, ….*"

**Answers to Anonymous Reviewer #2**

*"Near East Desertification: Impact of Dead Sea drying on convective rainfall" by Khodayar and Hoerner submitted to Atmospheric Chemistry and Physics*

Dear Reviewer2:
Thanks for your comments and suggestions. In the following you can find a more detail answer to your comments (in blue).
Kind regards
Samiro Khodayar
* * *
This is the 2nd round of reviews of the manuscript that presents an impact of Dead sea on the precipitation and evaporation in the surrounding areas. The authors have addressed satisfactorily my previous comments, but I still have a few specific comments. I list them below.

Specific comments:

1. Lines 51-54: Please reformulate the sentence. It is too long and is using "being" two times – sounds a bit strange.
"Since the Dead Sea is a terminal lake of the Dead Sea Valley, no natural outflow exists, evaporation is the main loss of water. The wind velocity and vapour pressure deficit are identified as the main governing factors of evaporation throughout the year (Metzger et al. 2017)."

2. Line 135: "... to have into..." -> "... to take into..."
Corrected

3. Line 160: "..., but with additional hourly output."
Corrected

4. Line 161: This is a bit strange line, and I would omit it.
Corrected

5. Lines 163-168: You already write that in the previous section. Only once is enough... either here or in the previous section.
Thanks for the observation, we removed these lines here.

6. Lines 175-192: Although I would prefer that you show the evaluation of the simulation as a subsection in the results (or even as supplementary information), I find this part here a bit out of blue. It is too detailed and too long for something that you do not show, and this is still the methodology section. In addition, you only focus on the precipitation, but later in the results, you also look into other fields. I wonder if this can be summarized in a few lines, something like: "Comparison of the simulation with observations shows better/improved results at higher resolution, especially for precipitation, although some biases exist. These biases occur over... and could be related to...(refs)"
Following the reviewer´s advice we have shortened this part.

7. Lines 265-267: This is a bit strange line and I think it needs a reformulation in order to properly present what you did.

We have tried to improve this explanation.

8. Line 298: Decrease of 0.5 % is quite small, but this is (I assume) for mean precipitation. Maybe you should write then that this is for the mean precipitation, but you could also estimate that number for heavy precipitation.

We have indicated this value is in relation to the mean areal and temporal precipitation. To avoid misleading information we did not include a percentage for HP since this will be highly dependent on the season and/or type of events considered.

9. Lines 321-324: Note that these are now negative differences, so please carefully revise this part and all others where this change has an impact on the results.

Thanks for noticing, we have carefully read again the text and made the appropriate changes.

10: Table 1: Again, the event of 14.11 is missing. Instead, there is 15.11. Are these the same events and why are they marked differently in the table and in the text?
Thanks for noticing, this is just a mistake, the whole event covers the 14 to 15.11. We have corrected this information.

11. Line 1118: SEN(14.11) – This should be 19.11... the same as REF.
Corrected

12. Lines 1148-1149: Again, wrong REF and SEN... It should be 19.11 and not 14.11.
Corrected

13. Lines 1163-1164: The same as comment 11.
Corrected

14: Regarding the general comment 4 from a previous revision and replies to Reviewer 1: You should then also mention the spin-up period in the methodology... how you define it and if you discard some years from the analysis.
The spin-up time selection was already mentioned in L173-174.

[revised manuscript text omitted]

---

## Author Response (AR3)

**Comments to the editor:**

- **The recommendation is that you avoid the term "Dead Sea drying out" entirely.**

The term "Dead Sea drying out" has been removed entirely in all places where it could lead to a misunderstanding.

**Specifically, I suggest the following changes prior to accepting your paper for publication in ACP:**

**1) Change the title to emphasise the idealized aspects of your study. One option might be: "An idealized model sensitivity study on Dead Sea desertification with a focus on the impact on convection"**

Thank you for the suggestion, the title has been changed using your suggestion which I consider very well represent the purpose of the paper clarifying the idealized-model representation of the Dead Sea in our simulations.

**2) Change key words**
Changed

**3) Change abstract (lines 5, 14 where you mention "drying out of Dead Sea" - reword these sentences and better emphasise the idealized aspect of your sensitivity experiment.**
Changes have been applied to the abstract following your suggestions. Instead of "drying out of Dead Sea" the concept of "changed conditions at the Dead Sea" has been used thought the paper and the idealized aspect of the sensitivity experiment in relation to the description of the Dead Sea characteristics has been explicitly exposed.

**4) Abstract line 21: this last sentence is very vague. It would be good to give here more quantitative information about how much evaporation and precipitation decrease, and air temperature increases (the same again on line 560). If I interpret Fig. 2a correctly, then the effect on precipitation is very weak.**
As you correctly mention the effect is very weak, as indicated previously in the manuscript about 0.5 %. This information has been included in the abstract and conclusions.

**5) Please reconsider / clarify figure captions. For instance, Fig. 2 mentions "areal-daily averaged ... evaporation ..." and Fig. 4 mentions "monthly-daily accumulated areal mean values of evaporation ...". I don't understand "monthly-daily"?? I think that Fig. 2 shows area-averaged daily accumulated values and Fig. 4 shows area-averaged monthly accumulated values. Please clarify.**
This information has been changed following the editor´s suggestion.

**6) Change conclusions and general wording of simulations: emphasise idealized setup in line**

**554; and change several statements in lines 558ff. You could maybe refer to the "drying of the Dead Sea" sensitivity experiment in the entire paper as the "bare soil everywhere simulation" or shorter "bare soil simulation" and the control run as the "Dead Sea simulation".**

This information has been modified throughout the paper following the editor´s suggestion.

**7) line 562: I don't understand how a cooling effect can lead to an increase of temperature?? Please rephrase this sentence.**

This sentence has been removed from this part of the text to avoid repetition since it has been described before in section 3.1.

**8) line 621: careful - I am not sure that your study tells much about the effect of "lake level decline". Lake level decline means that there is still water but at a lower level and with reduced surface area. But your idealized experiment is a complete bare soil experiment, which is something else than a decline of the lake level. It could be rewarding to do several intermediate experiments between today's conditions and bare soil only.**

The term "lake level decline" has been removed from the manuscript everywhere where it could lead to misunderstanding to the reader.

In follow-up experiments in the region, with improved description of the Dead Sea characteristics among other improvements, we are performing intermediate experiments, which we expect to publish in following publications.

[revised manuscript text omitted]